# MicroRNAs shape circadian hepatic gene expression on a transcriptome-wide scale

Ngoc-Hien Du[1†], Alaaddin Bulak Arpat[1,2†], Mara De Matos[1], David Gatfield[1]*

[1]Center for Integrative Genomics, University of Lausanne, Lausanne, Switzerland;
[2]Vital-IT, Swiss Institute of Bioinformatics, Lausanne, Switzerland

**Abstract** A considerable proportion of mammalian gene expression undergoes circadian oscillations. Post-transcriptional mechanisms likely make important contributions to mRNA abundance rhythms. We have investigated how microRNAs (miRNAs) contribute to core clock and clock-controlled gene expression using mice in which miRNA biogenesis can be inactivated in the liver. While the hepatic core clock was surprisingly resilient to miRNA loss, whole transcriptome sequencing uncovered widespread effects on clock output gene expression. Cyclic transcription paired with miRNA-mediated regulation was thus identified as a frequent phenomenon that affected up to 30% of the rhythmic transcriptome and served to post-transcriptionally adjust the phases and amplitudes of rhythmic mRNA accumulation. However, only few mRNA rhythms were actually generated by miRNAs. Overall, our study suggests that miRNAs function to adapt clock-driven gene expression to tissue-specific requirements. Finally, we pinpoint several miRNAs predicted to act as modulators of rhythmic transcripts, and identify rhythmic pathways particularly prone to miRNA regulation.

## Introduction

Circadian clocks orchestrate daily oscillations in mammalian behaviour, physiology, and gene expression. In mammals, a master pacemaker in the brain's suprachiasmatic nucleus (SCN) synchronises subsidiary oscillators present in most peripheral cell types (reviewed in *Dibner et al., 2010*; *Mohawk et al., 2012*). Timekeeping by peripheral and SCN clocks relies on negative transcriptional feedback loops that engender oscillatory gene expression. In the core loop, BMAL1:CLOCK transcription factor heterodimers drive the expression of repressors, encoded by the *Period (Per1,2)* and *Cryptochrome (Cry1,2)* genes. PER:CRY protein complexes subsequently accumulate in the nucleus and repress BMAL1:CLOCK-mediated transcription. Due to their instability, repressor protein and mRNA abundance rapidly drops below the threshold required for autorepression, clearing the way for a new cycle. The mechanisms that control the stability of clock proteins have been studied to considerable extent and frequently involve post-translational protein modifications that control proteasomal degradation (e.g., *Yagita et al., 2002*; *Eide et al., 2005*; *Shirogane et al., 2005*; reviewed in *Mehra et al., 2009*; *Chong et al., 2012*). It is less well understood how the decay of core clock mRNAs is controlled (reviewed in *Kojima et al., 2011*; *Lim and Allada, 2013*).

Cyclically expressed transcription factors such as BMAL1:CLOCK (*Panda et al., 2002*; *Rey et al., 2011*) or REV-ERBα/β (*Ueda et al., 2002*; *Le Martelot et al., 2009*; *Bugge et al., 2012*; *Cho et al., 2012*) relay the timing information from the core clock to clock output pathways by driving the rhythmic expression of clock-controlled genes (CCGs), many of which are tissue-specific (*Panda et al., 2002*; *Storch et al., 2002*). In mouse liver, up to 15% of expressed mRNAs accumulate in a rhythmic fashion (*Vollmers et al., 2009*; *Mohawk et al., 2012*). At least in part, the synergistic activation of genes by circadian and tissue-specific transcription factors may account for the rhythmic expression of cell type-specific transcripts. Conceivably, tissue-specific post-transcriptional regulation could participate in this endeavour as well. Recent studies in liver have indeed suggested that a substantial proportion of mRNA abundance rhythms is generated post-transcriptionally (*Koike et al., 2012*; *Le Martelot et al.,*

*For correspondence: david.gatfield@unil.ch

†These authors contributed equally to this work

Competing interests: The authors declare that no competing interests exist.

**eLife digest** The rising and setting of the sun have long driven the schedules of humans and other mammals. This 24-hr cycle influences many behavioural and physiological changes, including alertness, body temperature, and sleep. A region in the brain acts as a master clock that regulates these daily cycles, which are called circadian rhythms.

Signals from the brain's master clock turn on and off 'core clock genes' in cells, which trigger cycles that cause some proteins to be produced in a circadian rhythm. The rhythm is specialized to a particular tissue or organ, and may help them to carry out their designated daily tasks. However, circadian rhythms might also be produced in other ways that do not involve these genes.

Messenger RNA (mRNA) molecules have a central role in the production of proteins, and in the mouse liver, up to 15% of mRNA molecules are produced in circadian cycles. The liver performs essential tasks that control metabolism–including that of carbohydrates, fats, and cholesterol. Precisely timing when certain mRNAs and proteins reach peaks and troughs in their activities to coincide with mealtimes is important for nutrients to be properly processed.

Other RNA molecules called microRNAs influence how mRNA molecules are translated into proteins. Now Du, Arpat et al. have looked at the influence of microRNAs on circadian rhythms in the mouse liver in greater detail. These experiments, which involved 'knocking out' a gene that is essential for the production of microRNAs, show that rather than setting the mRNA rhythms, the microRNAs appear to adjust them to meet the specific needs of the liver. Targeting specific microRNA molecules may reveal new strategies to tweak these rhythms, which could help to improve conditions when metabolic functions go wrong.

2012; *Menet et al., 2012*) and have speculated on the involvement of miRNAs in this process (*Menet et al., 2012*).

MicroRNAs are short, non-coding RNA molecules that inhibit the translation and promote the destabilisation of mRNAs by base-pairing with sequence elements that are typically located in the 3′ untranslated regions (3′ UTRs) of their target transcripts (reviewed in *Krol et al., 2010*; *Fabian and Sonenberg, 2012*; *Yates et al., 2013*). Mammalian genomes encode >1000 miRNAs (*Bentwich et al., 2005*) and each may have hundreds of targets. It has thus been estimated that up to 60% of mammalian protein-coding transcripts can be regulated by miRNAs (*Lewis et al., 2005*; *Friedman et al., 2009*). It is therefore likely that these regulatory molecules carry functions in circadian gene expression as well, both at the level of the core clock and clock output genes. The antisense-inactivation of miR-219 and miR-132 in the SCN has indeed been reported to result in mild lengthening of period and in defective light-induced clock resetting, respectively (*Cheng et al., 2007*). Moreover, an elegant study by *Chen et al. (2013)* has recently reported dramatic period shortening (≈2 hr) of free-running rhythms in miRNA-deficient mouse embryonic fibroblasts (MEFs), likely caused by the lack of three miRNAs (miR-24, miR-29a, miR-30a) targeting *Per1* and *Per2* mRNAs. Other miRNAs have been noted for their capacity to regulate core clock transcripts as well (*Kiriakidou et al., 2004*; *Meng et al., 2006*; *Nagel et al., 2009*; *Tan et al., 2012*; *Lee et al., 2013*; *Shende et al., 2013*), but their circadian functions in vivo are still unclear. Similarly, the functions that miRNAs assume in clock output pathways have barely been investigated in mammals, except in a few cases. The *mir122* locus thus encodes a highly abundant, hepatocyte-specific miRNA involved in the regulation of cholesterol and lipid metabolism (*Krutzfeldt et al., 2005*; *Esau et al., 2006*) that is clock-controlled in mouse liver (*Gatfield et al., 2009*; *Menet et al., 2012*). Mature miR-122 appears to act as a modulator of circadian output pathways, despite being non-rhythmic due to its high metabolic stability (*Gatfield et al., 2009*; *Kojima et al., 2010*). Conceivably, other miRNAs could target rhythmic transcripts as well.

We have now comprehensively analysed the contribution that miRNAs make to the regulation of hepatic circadian gene expression by genetically inactivating miRNA biogenesis in the livers of adult mice. Around-the-clock profiling of mRNAs and pre-mRNAs from *Dicer*-deficient (*Harfe et al., 2005*) and control livers thus allowed the global detection of post-transcriptional and miRNA-dependent regulation of hepatic gene expression. We have addressed the following main questions. What functions do miRNAs have in the regulation of the hepatic core clock? To what extent do miRNAs contribute to the post-transcriptional generation of mRNA rhythms? And third, for transcripts subject to both miRNA

regulation and cyclic transcription, how do these mechanisms integrate and cooperate to yield the net rhythmic output? Finally, the comprehensive atlas of miRNA-dependent (circadian and non-circadian) gene expression established in this study exhaustively charts the regulatory space that miRNAs occupy in liver gene expression and may thus serve as an important resource beyond chronobiology.

## Results

### Hepatocyte-specific, conditional *Dicer* inactivation allows for the disruption of miRNA biogenesis in the livers of adult mice

To comprehensively uncover miRNA-regulated gene expression in the liver, we generated a mouse model in which miRNA biogenesis could be inactivated in hepatocytes, which constitute around 80% of liver mass (*Weibel et al., 1969*). We used mice carrying conditional knockout alleles for the *Dicer1* gene (termed *Dicer^flox* in the following), which encodes the ribonuclease that converts pre-miRNAs to mature miRNAs (*Harfe et al., 2005*), and a *Cre-ER^T2* recombinase expressed as an internal ribosome entry site (IRES)-controlled transgene in the 3' UTR of the endogenous *Albumin* locus (*Alb^Cre-ERT2* in the following) (*Schuler et al., 2004*). The *Albumin* locus conferred hepatocyte-specificity and the Cre-ER^T2 fusion allowed for induction of knockout in adult animals using tamoxifen.

Previous publications have reported constitutive hepatocyte-specific knockouts of *Dicer* (*Hand et al., 2009*; *Sekine et al., 2009a, 2009b*) and to our knowledge the alleles we employed have not been used in combination before. We thus first confirmed their suitability for efficient miRNA depletion in the livers of adult animals. To this end, we analysed the kinetics of recombination at the *Dicer* locus at different timepoints after intraperitoneal tamoxifen (tx) injection. Recombination was undetectable before but highly efficient from 2 to 3 weeks after tx treatment (*Figure 1—figure supplement 1A*). We assessed the consequences on miRNA expression by northern blot analysis probing for the highly abundant, hepatocyte-specific miR-122, which has been previously noted for its long half-life (*Gatfield et al., 2009*) and thus served as a proxy for overall miRNA depletion from hepatocytes. One month after tx treatment, miR-122 was virtually undetectable (*Figure 1—figure supplement 1B,C*). Moreover, loss of mature miR-122 was accompanied by increased levels of its precursor, pre-miR-122 (*Figure 1—figure supplement 1B,C*), as expected (*Harfe et al., 2005*). Interestingly, we observed that at later timepoints beyond 2 months after tx treatment, miR-122 expression began to recover (*Figure 1—figure supplement 1D,E*). The most likely explanation for this observation was that after miRNA loss the liver eventually renewed its hepatocytes from a cell population that had escaped recombination. In summary, we concluded that our experimental system appeared suitable for the efficient knockout of *Dicer* and, at least judging by miR-122, for the depletion of miRNAs. Approximately 1 month after tx treatment appeared to represent a suitable timepoint for experiments aiming at a comparison of knockout with control animals.

We induced the knockout in more than 40 mice (male, aged 3–6 months), entrained them to light–dark cycles for 1 month (LD 12:12; *ad libitum* feeding) and sacrificed them at 4-hr intervals around-the-clock (*Figure 1A*). A littermate control group was treated identically. We combined both wild-type and heterozygous males for controls because a single functional *Dicer* allele was sufficient to ensure miRNA processing to wild-type levels (*Figure 1—figure supplement 2A*), as expected from previous studies (*Harfe et al., 2005*; *Kanellopoulou et al., 2005*; *Murchison et al., 2005*; *Chen et al., 2008*; *Frezzetti et al., 2011*). Pools of liver total RNA (3–4 animals) were assembled into two independent full time series around-the-clock (ZT0-20) for both knockouts and controls. Northern blot analysis confirmed loss of miR-122 in the knockouts across all timepoints (*Figure 1B,C*). Microarray (*Figure 1—figure supplement 2B*) and northern blot analyses (*Figure 1—figure supplement 2C*) showed that beyond miR-122, miRNA levels were globally decreased. Other hepatocyte-enriched miRNAs (e.g., miR-148a, miR-192 and miR-194 [*Sun et al., 2004*; *Landgraf et al., 2007*; *Tang et al., 2007*; *Farid et al., 2012*]) were thus virtually undetectable in knockouts. Probably owing to their expression also in non-hepatocyte cells, known ubiquitous miRNAs (e.g., miR-21 [*Landgraf et al., 2007*], miR-26a, or let-7 family members [*Lagos-Quintana et al., 2003*]) decreased less strongly, and miRNAs with a particularly strong expression in non-hepatocytes (e.g., miR-126 in hepatic stellate cells [*Guo et al., 2013*]) were almost unchanged, as expected (*Figure 1—figure supplement 2B,C*). In all cases, reduced miRNA levels correlated with increased pre-miRNA abundance (*Figure 1B,D*, *Figure 1—figure supplement 1C*). Moreover, even in the knockout, pre-miR-122 still showed rhythmic accumulation with a trough at ZT12 (*Figure 1B,D*), which was a consequence of clock-driven rhythmic transcription at the *mir122* gene locus (*Gatfield et al., 2009*;

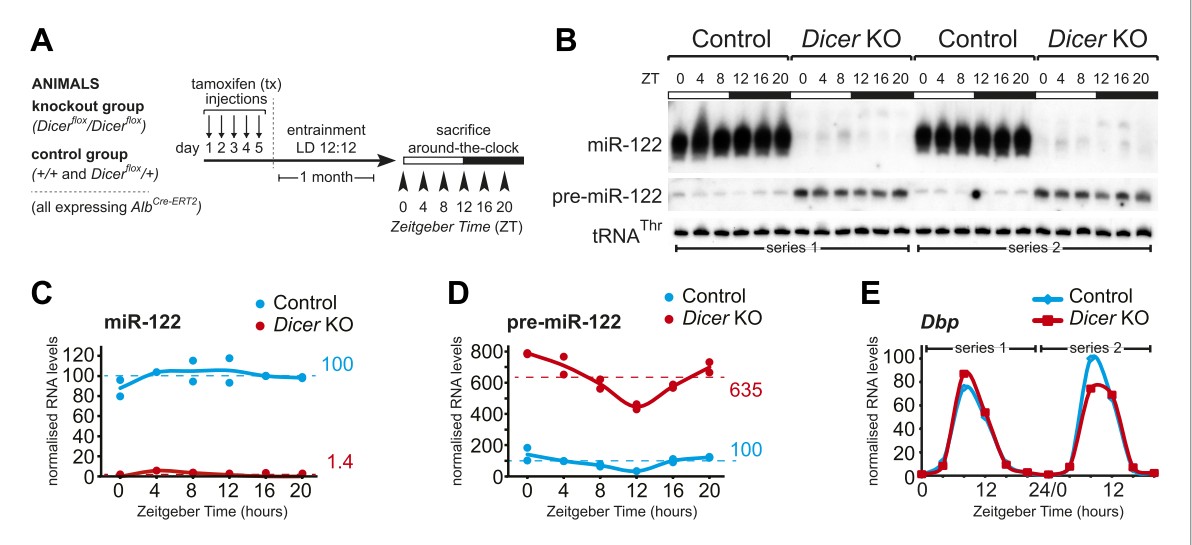

**Figure 1**. Analysis of hepatic *Dicer* knockout using miR-122 as a diagnostic marker. (**A**) Schematic of the *Dicer* knockout protocol used throughout the study. Conditional knockout and control littermates (heterozygotes and wild-type) carrying the *Alb^Cre–ERT2* allele that ensures hepatocyte-specific expression were injected with tamoxifen, entrained to 12-hr light/12-hr dark cycles for a month, and sacrificed at the indicated *Zeitgeber* Times. (**B**) Northern blot analysis demonstrating that miR-122 is virtually undetectable in knockout animals. In contrast, the Dicer substrate, pre-miR-122, accumulates to higher levels due the lack in turnover by Dicer. Each loaded sample is a mix of RNAs prepared from 3 to 4 independent animals. The two series are the same as used for the RNA-seq analysis. tRNA^Thr served as a loading control. (**C**) Quantification of miR-122 abundance from northern blot shown in (**B**) after normalisation to tRNA^Thr. In the knockouts, mean miR-122 levels only reached 1.4% of control levels. (**D**) Quantification of pre-miR-122 expression (normalised to tRNA^Thr) from northern blot shown in (**B**). On average, pre-miR-122 accumulates to >sixfold higher levels than in controls due to the absence of Dicer processing. Note that even in the knockout, pre-miR-122 still shows the rhythmic accumulation that is the result of cyclic transcription at the *mir122* locus. (**E**) Quantitative real-time PCR analysis of *Dbp*, a typical core clock output gene, clearly indicates that rhythmic gene expression per se still occurs in *Dicer* knockout livers.

The following figure supplements are available for figure 1:

**Figure supplement 1**. Kinetics of *Dicer* knockout in liver.

**Figure supplement 2**. Global analysis of miRNA depletion in *Dicer* knockout livers.

*Menet et al., 2012*). Other clock-controlled genes such as *Dbp* (*Ripperger et al., 2000*) still oscillated as well (*Figure 1E*). We thus concluded that cyclic gene expression per se was still functional in *Dicer* knockouts and that the RNA time series collected from the two genotypes was suitable to uncover the effects of miRNA loss on the rhythmic expression of core clock and clock output genes in hepatocytes.

## RNA-seq of pre-mRNAs and mRNAs distinguishes transcriptional from post-transcriptional regulation

MicroRNAs inhibit the translation and promote the decay of their mRNA targets. How these events are elicited, and in which order they occur, is still subject of ongoing debate. However, given that in most cases mRNA degradation appears to be an endpoint in miRNA activity, changes in mRNA levels are the most commonly used readout for regulation by miRNAs (reviewed in *Fabian and Sonenberg, 2012*).

To analyse how rhythmic RNA accumulation was affected by the loss of miRNAs, we employed RNA high-throughput sequencing (RNA-seq). We chose a protocol for library generation that was based on the random priming of rRNA-depleted ('Ribo-Zero') total RNA, which permitted the simultaneous quantification of mRNAs (exon-mapping reads) and pre-mRNAs (intron-mapping reads). Given that splicing occurs co-transcriptionally and is relatively fast, the intron-containing pre-mRNAs are short-lived and their abundance can serve as a proxy for gene transcription rates (*Koike et al., 2012*). This strategy thus identifies post-transcriptional (mRNA) gene expression changes that occur in the *Dicer*

knockouts and likely represent bona fide miRNA targets, as opposed to transcriptional changes (mRNA and pre-mRNA affected), which likely represent secondary effects of miRNA loss. Conceivably, any transcription factor that is a direct miRNA target would engender numerous such secondary effects.

We sequenced knockout and control time series with comparably high coverage. Reads mapped to different RNA classes, as expected (*Figure 2A*, *Figure 2—figure supplement 1*, *Figure 2—source data 1*). We did not detect gross distortions in RNA populations that would compromise the comparability of knockout and control samples, although some differences reached statistical significance (*Figure 2—figure supplement 1*, *Figure 2—source data 1*). We then examined transcripts from protein-coding loci, which contributed around half of all obtained sequences (51.6 ± 1.0% in knockout, 47.9 ± 1.3% in control; *Figure 2A,B*, *Figure 2—figure supplement 1*, *Figure 2—source data 1*). Reads mapping to exonic sequences were categorised as 'mRNA' and those mapping to introns or intron–exon boundaries as 'pre-mRNA' ('Materials and methods'). Pre-mRNAs are of generally low abundance, but genome-wide there is an estimated 20-fold more transcribed intronic than exonic sequence space (*Sakharkar et al., 2005*). In accordance, pre-mRNAs were readily detectable transcriptome-wide, representing around a quarter of protein-coding reads (25.5 ± 4.1% in knockout, 21.5 ± 3.1% in control; *Figure 2A,B*, *Figure 2—source data 1*). Finally, the RNA-seq data also provided an additional confirmation of the high efficiency of the *Dicer* knockout (*Figure 2—figure supplement 2*).

In the complete data set, we detected 13,408 individual protein-coding loci (*Figure 2C*). The intersecting set (mRNA and pre-mRNA detected in knockout and control) contained 11,990 genes (89.4%); another 361 transcripts (2.7%) common to both genotypes corresponded to intronless genes. To assess the changes engendered by miRNA loss, we first compared mRNA and pre-mRNA levels separately between the genotypes (averaging over timepoints). It was apparent that *Dicer* deficiency caused large changes in mRNA and pre-mRNA abundance (*Figure 2D,E*). Applying a 1.5-fold cut-off on differential expression, we found that 30.3% of mRNAs and 15.5% of pre-mRNAs were significantly changed in the *Dicer* knockout. These changes were skewed towards increased levels for mRNAs (19.8% increased, 10.5% decreased) but slightly less so for pre-mRNAs (9.8% increased, 5.7% decreased). In summary, these findings confirmed our expectation that global miRNA loss would lead to the widespread de-repression of targets that affected a substantial part of the transcriptome. Inevitably, these primary effects would provoke numerous secondary responses involving altered transcription. Striking transcriptional effects were detected for several imprinted loci and for a group of known development-specific and sexually dimorphic genes (see *Figure 2—figure supplement 3* for examples), as previously reported from *Dicer* knockouts (*Hand et al., 2009*; *Sekine et al., 2009a*). We excluded the group of transcriptionally most altered genes in subsequent circadian analyses, as vastly different transcription rates would likely confound comparative analyses of rhythmic properties, in particular regarding the precise post-transcriptional contributions (marked in red in *Figure 2D,E*).

We next calculated transcriptome-wide mRNA/pre-mRNA ratios, which are a measure of mRNA stability and thus an indicator of post-transcriptional regulation (*Zeisel et al., 2011*). The distribution of mRNA/pre-mRNA ratios was globally shifted to higher values in the knockout (*Figure 2F*) and we thus assessed whether an increased mRNA/pre-mRNA ratio in the *Dicer* knockout could be predictive of miRNA activity. Using experimentally validated targets (*Hsu et al., 2011*) of miR-122 as a positive control and of miR-124 as a negative control (this miRNA is not expressed in liver, [*Landgraf et al., 2007*]), we indeed observed that the majority of miR-122 targets showed increased mRNA/pre-mRNA ratios in *Dicer* knockouts, whereas most miR-124 targets did not (*Figure 2G*). Ribosomal protein mRNAs, whose 3' UTRs are particularly short (*Caldarola et al., 2009*) and thus probably not regulated by miRNAs, did not show increased mRNA/pre-mRNA ratios either (*Figure 2G*). We thus concluded that incorporating mRNA/pre-mRNA ratios in our analyses could enrich for likely direct miRNA targets in the dataset.

## The liver core clock is relatively resilient to miRNA loss

We next assessed the expression of transcripts encoding core clock components and clock regulators (listed in *Figure 3—source data 1*). With regard to mRNA/pre-mRNA ratios, the transcripts of the *Period* gene family (*Per1, Per2, Per3*) showed a moderate (average 1.6-fold), but significant increase in *Dicer* knockouts (*Figure 3A*). The time-resolved RNA-seq data (*Figure 3B*) confirmed that *Per1* and *Per2* mRNAs accumulated to higher levels throughout the day despite similar rhythmic pre-mRNA abundance. For *Per3*, the higher mRNA/pre-mRNA ratio mainly resulted from decreased transcription (*Figure 3—figure supplement 1*). Moreover, we noticed that *Cry2* showed post-transcriptional

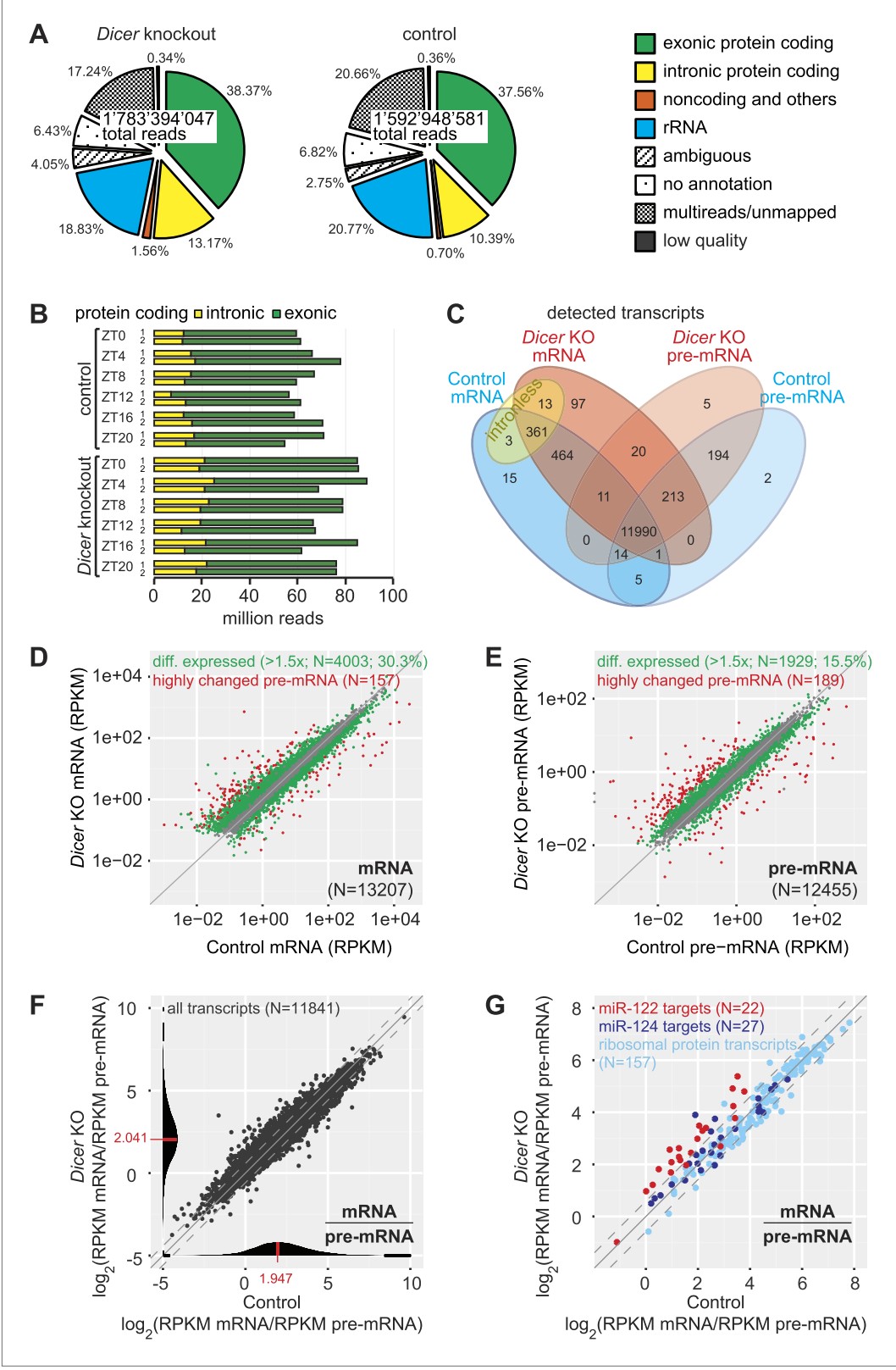

**Figure 2**. Genome-wide quantification of pre-mRNA and mRNA abundance by RNA-seq. (**A**) Summary of RNA-seq results from *Dicer* KO (left) and control (right). Percentages are relative to total reads across all time points. (**B**) Time-resolved analysis of intron- (yellow) and exon-mapping (green) RNA-seq reads in the two series of *Figure 2. Continued on next page*

*Figure 2. Continued*

knockout and control mice. (**C**) Venn diagram indicating the number of genes whose expression was detectable in *Dicer* knockout and control animals on the mRNA (exon) and pre-mRNA (intron) levels (threshold: 0.1 RPKM for mRNA, 0.01 RPKM for pre-mRNA in at least 1/3 of samples). (**D**) Comparison of mRNA expression (RPKM) of protein coding genes in *Dicer* knockouts vs controls (averaged over all time points). Green dots correspond to transcripts whose levels are statistically significantly >1.5-fold different between genotypes and considered as differentially expressed. The red mRNAs are highly different on the transcriptional (pre-mRNA) level and were excluded from circadian analyses; see (**E**). (**E**) Comparison of pre-mRNA expression (RPKM) of protein coding genes in *Dicer* knockouts vs controls. Green dots correspond to transcripts whose levels are statistically significantly >1.5-fold different between genotypes and considered as differentially expressed. Red dots correspond to genes with higher pre-mRNA fold-change than the 90% quantile of differentially expressed pre-mRNAs (corresponding to >4.1-fold expression differences between knockout and control); due to the highly different transcription rates, they were excluded from circadian analyses. (**F**) Analysis of transcriptome-wide mRNA/pre-mRNA ratios (as a measure of mRNA stability) between *Dicer* knockouts and controls. Note that in the *Dicer* KO, mRNAs become globally more stable (modes of distributions indicated in red). (**G**) mRNA/pre-mRNA ratio changes are predictive for direct miRNA targets. Known targets of liver-specific miR-122 (red) thus have overall increased mRNA/pre-mRNA ratios in the knockout, in contrast to targets of miR-124 (dark blue), which is not expressed in liver. Transcripts encoding ribosomal proteins (pale blue) seem to be overall excluded from miRNA regulation, as expected.

The following source data and figure supplements are available for figure 2:

**Source data 1**.

**Figure supplement 1**. Analysis of RNA-seq data by RNA classes.

**Figure supplement 2**. Reads mapping to the *Dicer* locus confirm high knockout efficiency.

**Figure supplement 3**. Several genes show extreme transcriptional changes in *Dicer* knockouts.

up-regulation in the *Dicer* knockout and was thus a potential miRNA target, although the case was less clear than for *Per1* or *Per2* since *Cry2* transcription was increased at certain timepoints as well. Beyond these genes, the expression of most other core clock components, including *Cry1*, *Clock*, *Bmal1* (*Figure 3B*) and others (*Figure 3—figure supplement 1*) only showed minor changes, if at all. Consistent with the mRNA profiles, western blot analysis confirmed an estimated twofold increase in the levels of PER2 protein and 1.5-fold higher CRY2 (*Figure 3C*). Surprisingly, PER1 levels were barely increased (1.1-fold higher). A number of mechanisms could account for the observed uncoupling of protein from mRNA levels, including secondary effects of miRNA loss acting on *Per1* mRNA translation or the protein degradation machinery. While we cannot pinpoint the exact mechanism that is operative in counter-regulating *Per1* mRNA increase, it is interesting to note that the RNA binding protein *Syncrip* (also known as *hnrnp q*) is a positive regulator of *Per1* translation (*Lee et al., 2012*) and showed decreased expression in *Dicer* knockouts (*Figure 3—figure supplement 1*). Importantly, however, it was clear that core clock functionality was overall unimpaired, as also exemplified by the virtually unchanged expression profiles of genes that are directly regulated by CLOCK:BMAL1, PERs and CRYs, such as *Dbp* (*Figure 3—figure supplement 1*, *Figure 1E*).

In our knockout model, miRNA loss occurred in the hepatocytes of a liver that was under constant entrainment by a genetically intact SCN clock. It was thus conceivable that a more severe phenotype was masked and would only become apparent under non-entrained, free-running conditions. It has indeed recently been reported that the period length of free-running rhythms in *Dicer*–deficient MEFs is dramatically shortened by ≈2 hr (*Chen et al., 2013*). Moreover, the authors identified the lack of miRNA-mediated repression of *Per1* and *Per2* as the underlying mechanism. We thus measured free-running rhythms in liver explants from *Dicer* knockout and control mice carrying the *mPer2^Luc* reporter allele (*Yoo et al., 2004*). Interestingly, our experiments showed a trend towards period lengthening in knockout liver explants with a period that was on average 41 min longer than in controls. The effect, however, did not reach statistical significance (p=0.197) (*Figure 3D,E*). Overall, we concluded that *Per1*, *Per2* and *Cry2* were likely miRNA targets in liver, but that the hepatocyte clock was relatively resilient to miRNA loss. The difference in phenotypes of liver and MEF *Dicer* knockouts could point to tissue-specific miRNA functions in clock regulation ('Discussion').

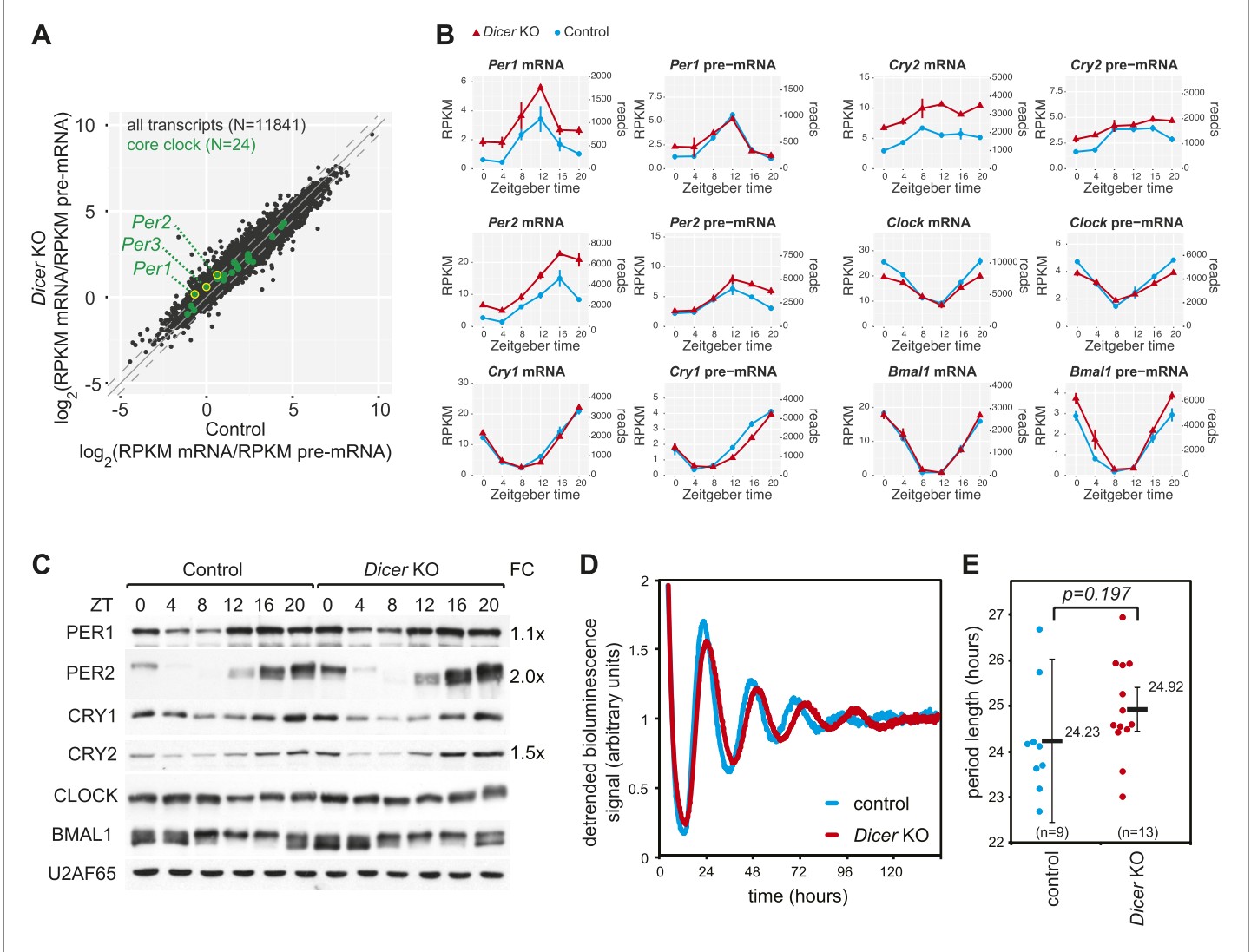

**Figure 3**. The hepatic core clock is remarkably resilient to miRNA loss. (**A**) mRNA/pre-mRNA ratio analysis of core clock transcripts indicates that *Per1*, *Per2*, and *Per3* have a significantly increased ratio in *Dicer* knockouts. (**B**) Time-resolved RNA-seq data for selected core clock transcripts with *Dicer* knockouts (red), controls (blue), mRNA (left panels), pre-mRNA (right panels), and the two time series plotted together in the same graph (vertical lines connect the data points from the two series). Abscissas show *Zeitgeber* Time, left ordinates RPKM values, right ordinates raw read numbers only normalised to sequencing depth in the samples. *Per1, Per2,* and *Cry2* thus show post-transcriptional upregulation in the *Dicer* KO, but other core clock transcripts such as *Clock, Bmal1* and *Cry1* do not. (**C**) Western blot analysis of core clock protein expression in liver nuclear extracts. For each sample, liver nuclear extracts from three mice were pooled. U2AF65 served as a loading control. Quantfication revealed that in the knockout the fold-change (FC) of up-regulation was ca. 2x for PER2 protein, 1.5x for CRY2, and 1.1x for PER1. Overall, the core clock appears to be fully functional. (**D**) Example of free-running rhythms measured from liver explants of *Dicer* knockout and control animals carrying the *mPer2^Luc* reporter gene. In these mice, a PER2-LUCIFERASE fusion protein is expressed from the endogenous *Per2* locus and allows for the real-time recording of circadian bioluminescence rhythms; importantly, the *mPer2^Luc* 3' UTR is identical to that of wild-type *Per2*. Raw bioluminescence was detrended using a 24-hr moving average. In this example, the knockout has a clearly longer period than the control. (**E**) Summary of several experiments as in (**D**) from a total of 9 control and 13 knockout mice. Although there is a trend to period lengthening upon miRNA loss, this effect is statistically not significant (p=0.197; student's *t* test).

The following source data and figure supplements are available for figure 3:

**Source data 1**.

**Figure supplement 1**. Core clock gene expression in *Dicer* knockouts.

# miRNAs drive the post-transcriptional cycling of a small number of mRNAs

We next explored how circadian gene expression was affected beyond the core clock. We first wished to identify transcripts whose rhythms were post-transcriptionally driven by miRNAs and assess whether this mechanism could underlie the discrepancies between cyclic transcription and cyclic mRNA accumulation that have been reported (*Koike et al., 2012*; *Le Martelot et al., 2012*; *Menet et al., 2012*). To identify oscillating gene expression transcriptome-wide and as comprehensively as possible, we adapted previously published methods for rhythmicity detection that we used at relatively low stringency ('Materials and methods'). Especially for pre-mRNAs, which are naturally noisier due to their low abundance, we wanted to avoid applying overly stringent criteria, as this could lead to the under-detection of transcriptionally and the overestimation of post-transcriptionally generated rhythms. Indeed, a previous study had noted that many genes that had failed to pass the imposed rhythmicity criteria on the transcriptional level visibly still exhibited transcriptional patterns that resembled those of the mRNA, albeit with much higher variability (*Menet et al., 2012*).

In control and *Dicer* knockout mice, 1630 and 1902 mRNAs, respectively, cycled with at least 1.5-fold peak-to-trough amplitudes (≈13% and 14%, respectively, of detected mRNAs; around half were shared between both groups; *Figure 4A*, *Figure 4—source data 1*), consistent with previous estimates that up to 15% of mRNAs are rhythmic in wild-type mice (*Lowrey and Takahashi, 2004*; *Vollmers et al., 2009*). On the pre-mRNA level, around 17% of transcripts were rhythmic (2083 in control and 2101 in knockouts; >50% shared). In both knockouts and controls, >60% of oscillating mRNAs were also rhythmic at the pre-mRNA level. These values are within the range of what has been reported previously from wild-type mice (*Koike et al., 2012*; *Le Martelot et al., 2012*; *Menet et al., 2012*). Interestingly, a sizeable number of transcripts were rhythmic specifically in either controls or *Dicer* knockouts (*Figure 4A*, *Figure 4—figure supplement 1*). We were particularly intrigued by the relatively high number of transcripts that appeared to become rhythmic in the *Dicer* knockouts (*Figure 4A*, sectors B–D) and investigated whether this was caused as a side effect of low stringency rhythm detection. However, the overlap of rhythmic transcripts between genotypes did not increase with higher thresholds despite reducing the overall number of detected rhythmic events (*Figure 4—figure supplement 2*). Further analyses revealed that *Dicer* KO-specific rhythmic transcripts were strongly enriched for certain Gene Ontology (GO) terms, of which cell cycle-specific genes (especially those involved in DNA replication, such as the members of the minichromosome maintenance complex, MCM) were particularly noteworthy (*Figure 4—source data 2*, see *Figure 4—figure supplement 3* for examples). These findings are in line with increased liver regeneration after miRNA loss (*Figure 1—figure supplement 1D,E*) and the circadian gating of the cell cycle in the regenerating liver (*Matsuo et al., 2003*).

To explore the relationship between transcript rhythmicity and miRNA regulation, we first consulted the mRNA/pre-mRNA ratios as an indicator of post-transcriptional regulation and of mRNA stability changes that could be indicative of direct miRNA activity (*Figure 4B,C*). For 30% of rhythmic transcripts in control mice (483 out of 1630), mRNA/pre-mRNA ratios were significantly increased in *Dicer* knockouts (*Figure 4—figure supplement 1A*, sectors I–P). Globally, rhythmic transcripts showed a slightly stronger shift to higher mRNA/pre-mRNA ratios than average (*Figure 4C*; p=0.001; Welch's two-sample *t* test). We concluded that several hundred circadian mRNAs were potentially miRNA-regulated but that there was no striking enrichment or depletion for miRNA regulation among the circadian transcriptome.

To identify candidates for post-transcriptionally driven rhythmicity involving miRNAs, we investigated the group of 290 transcripts whose cyclic accumulation occurred exclusively at the mRNA level in control animals (sector I in *Figure 4A*, see *Figure 4—source data 1*). A heatmap representation confirmed pronounced rhythmicity mainly at the control mRNA level, as expected, but also suggested that a considerable number of transcripts possibly still possessed underlying, yet noisier rhythms at the level of control pre-mRNA and/or *Dicer* knockout mRNA (*Figure 4D*). We thus examined all profiles individually, also taking into account mRNA/pre-mRNA ratio changes occurring in the *Dicer* knockouts (in total 93 of the 290 transcripts in sector I thus had increased mRNA/pre-mRNA ratios in knockouts, see *Figure 4—figure supplement 1*). The visual inspection confirmed that only a minority of transcripts showed the features expected for true miRNA-driven post-transcriptional rhythmicity and that, instead, many of the non-rhythmic assignments were probably a result of noisier data (Supplementary file 1 in *Du et al., 2014* and *Figure 4—source data 1* for whole transcriptome gene expression plots

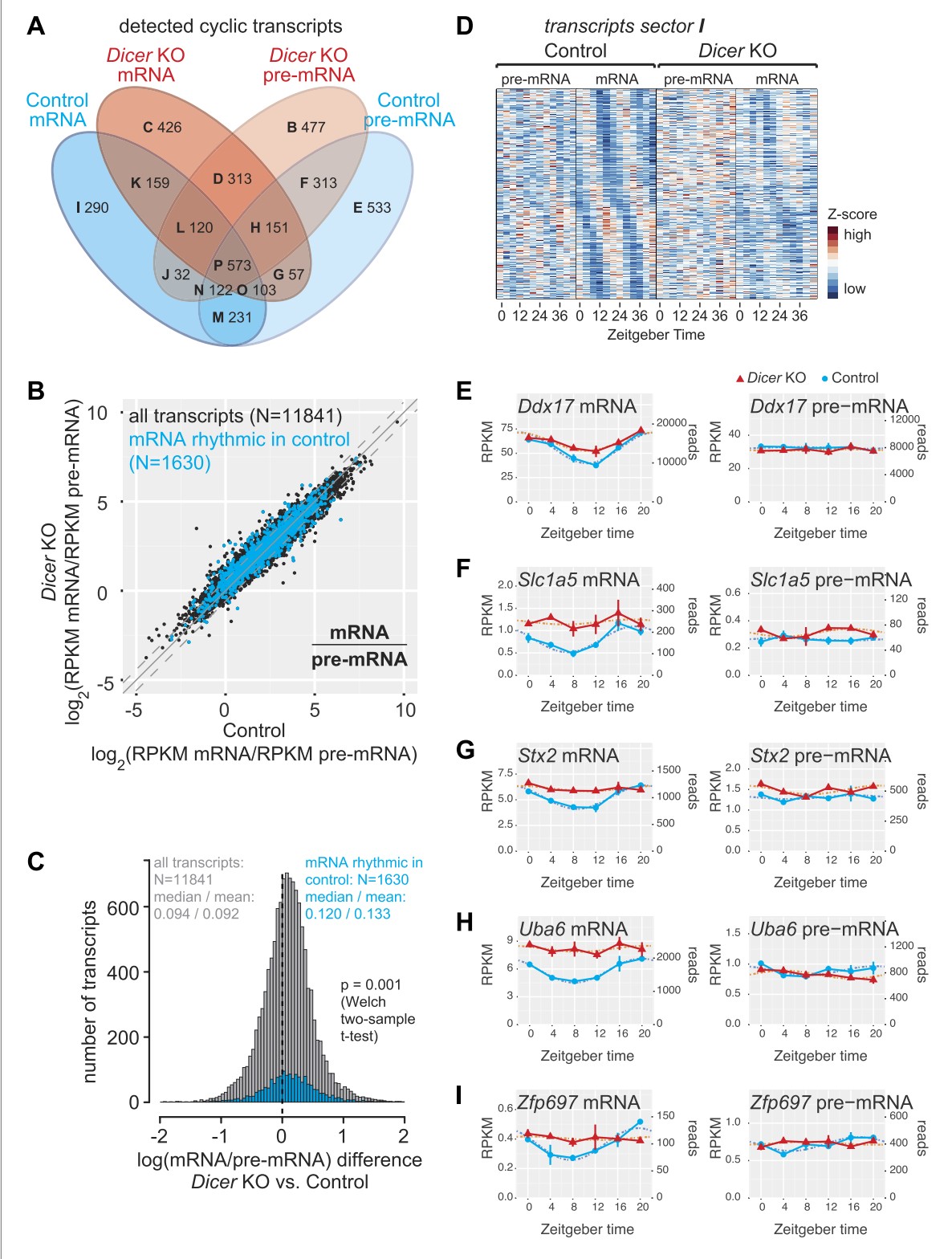

**Figure 4**. miRNAs may drive the rhythmic accumulation of a small set of transcripts. (**A**) Venn diagram summarising the extent of rhythmicity detected on all levels, that is mRNA and pre-mRNA in *Dicer* KO and control. A 1.5-fold threshold on peak-to-trough ratio amplitudes estimated from cosine fits was imposed. Sectors are named B–I for further reference. Sector A (not shown) corresponds to transcripts that were not rhythmic on any level. (**B**) mRNA/pre-mRNA ratio analysis for all transcripts that were detected as rhythmic in control (N = 1630). Globally, rhythmic transcripts thus appear to distribute

*Figure 4. Continued on next page*

*Figure 4. Continued*

similarly to all transcripts. (**C**) The distribution of the change in mRNA/pre-mRNA ratio between *Dicer* knockout and control is globally similar for rhythmic transcripts as compared to all transcripts. Although there is a slight statistically significant upshift in the cyclic transcripts (p=0.001, Welch two-sample *t* test), rhythmic transcripts do not seem to be specifically enriched for or excluded from miRNA regulation. (**D**) Heatmap of sector I transcripts, for which rhythmicity is only detected on the level of the mRNA in control animals. Note that although rhythms are clearly most pronounced for control mRNAs, as expected, the visual impression is that there are still underlying, but noisier rhythms present on the pre-mRNA level and in *Dicer* knockouts. pre-mRNAs (knockout and control) are on a common scale, as are the mRNAs. (**E–I**) *Ddx17, Slc1a5, Stx2, Uba6* and *Zpf697* are examples of transcripts that are transcriptionally non-rhythmic, but show mRNA rhythms that are *Dicer*-dependent, indicating that miRNAs could be involved in driving their post-transcriptional cycling. Note that all examples have relatively shallow rhythmic amplitudes. Interestingly, their rhythms fall into a similar phase, which could indicate that a common (rhythmic) miRNA could be involved. *Dicer* knockouts are shown in red, control in blue, mRNAs are on panels to the left, pre-mRNAs to the right; the two time series are plotted together in the same graph (vertical lines connect the data points from the two series). Abscissas show *Zeitgeber* Time, left ordinates RPKM values, right ordinates raw read number only normalised to sequencing depth in the samples. Dotted lines are the cosine fits to the data.

The following source data and figure supplements are available for figure 4:

**Source data 1**.

**Source data 2**.

**Figure supplement 1**. Transcriptome-wide rhythmicity detection in *Dicer* knockouts and controls.

**Figure supplement 2**. Differences in detected rhythmic transcripts between *Dicer* knockout and control are not caused by too low stringency rhythm detection.

**Figure supplement 3**. The expression of genes linked to the cell cycle/DNA replication becomes rhythmic in *Dicer* knockout livers.

**Figure supplement 4**. miRNAs as potential drivers of rhythmic mRNA accumulation.

**Figure supplement 5**. Quantitative real-time PCR analysis of several transcripts shown in main *Figures 4* and *5*.

and rhythmicity parameters). Visual selection resulted in no more than 20 transcripts for which we would confidently postulate that their rhythmic accumulation could indeed be miRNA-driven. As shown for *Ddx17, Slc1a5, Stx2, Uba6, Zfp697* (*Figure 4E–I*) and others (*Figure 4—figure supplement 4*), these transcripts were all characterised by relatively low amplitude rhythms. As a validation with an independent technique, we confirmed the circadian expression detected by RNA-seq for two of the genes, *Stx2* and *Uba6*, also by quantitative real-time PCR (qPCR) (*Figure 4—figure supplement 5A*). In summary, these findings suggested that miRNAs are only of marginal importance as drivers of circadian rhythmicity. Although we were able to identify several transcripts whose cyclic accumulation was *Dicer*-dependent and occurred post-transcriptionally, miRNA-mediated mechanisms can probably not account for the previously observed large discrepancy between mRNA and transcription rhythms (*Koike et al., 2012*; *Le Martelot et al., 2012*; *Menet et al., 2012*).

## miRNAs set the phase relationship between rhythmic transcription and rhythmic mRNA accumulation and modulate the cycling amplitudes

Hundreds of rhythmic mRNAs displayed increased stability (as judged from mRNA/pre-mRNA ratios) in *Dicer* knockouts (*Figure 4B,C*) and many of these transcripts likely represented direct miRNA targets. Circadian transcription paired with miRNA-mediated post-transcriptional control is thus probably a widespread phenomenon. We therefore investigated how such dual regulation added up at the level of mRNA cycling. Kinetic models of how mRNA stability influences rhythmic mRNA accumulation show that for a cyclically transcribed gene, the more stable the transcript, the later the phase and the lower the amplitude of its cycling (*Le Martelot et al., 2012*), as schematically shown in *Figure 5A*.

We examined the circadian properties of transcripts that were rhythmic across all conditions, that is in knockouts and controls at the pre-mRNA (>1.5-fold amplitude) and mRNA levels (without amplitude cut-off). Moreover, we grouped the transcripts according to whether their mRNA/pre-mRNA ratios were significantly increased in the *Dicer* knockout (i.e., transcripts more likely to represent direct miRNA targets; N = 167) or not (i.e., transcripts less likely to represent direct miRNA targets; N = 505).

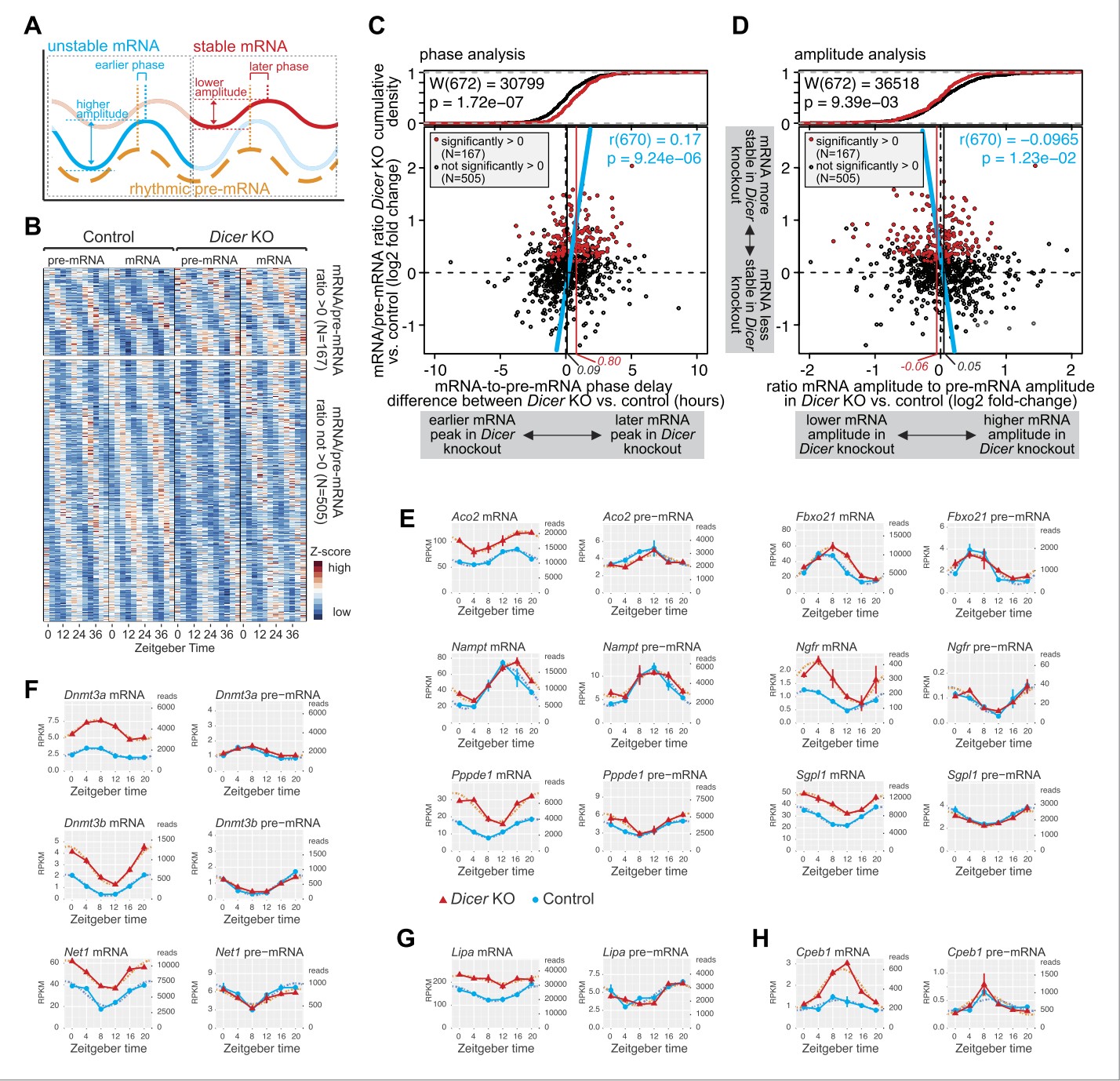

**Figure 5**. miRNAs adjust the mRNA phases and amplitudes of rhythmically transcribed genes. (**A**) Schematic representation of how mRNA stability affects circadian transcript accumulation. mRNAs with short half-lives (blue) will thus peak relatively early after the transcriptional peak (orange) and show higher peak-to-trough amplitudes than mRNAs with long half-lives (red). (**B**) Heatmap representation of analysed transcripts that all show rhythmic transcription (>1.5-fold amplitude) and rhythmic mRNA accumulation (no amplitude cut-off, in order to avoid biasing against transcripts whose amplitudes are regulated by miRNAs) in *Dicer* knockouts and controls. The top and lower panels show the transcripts with (N = 167) and without (N = 505), respectively, significantly increased mRNA/pre-mRNA ratios in *Dicer* knockouts. (**C**) Analysis of how the *Dicer* knockout changes the phases of rhythmic mRNA accumulation. Rhythmic transcripts from (**B**) are plotted according to the difference of the phase delay between mRNA and pre-mRNA peak in *Dicer* knockout vs control (abscissa; the further to the right a transcript is located, the later its mRNA is shifted in *Dicer* knockouts) and according to the change in mRNA/pre-mRNA ratio (ordinate; transcripts at the top of the panel thus become more stable in *Dicer* knockouts). Transcripts with significantly higher mRNA/pre-mRNA ratios in *Dicer* knockouts are marked in red. Red and black vertical lines correspond to mean phase shifts for the two groups of transcripts. The blue line shows the correlation of phase delay and mRNA/pre-mRNA ratio over all transcripts (Pearson's r(670) = 0.17; p=9.24e−06).

*Figure 5. Continued on next page*

*Figure 5. Continued*

Difference in phase delays between the two groups of transcripts was tested using Wilcoxon rank sum test (W(672) = 30,799; p=1.72e−07). At the top of the graph, the cumulative density plot shows that the two groups of transcripts are clearly separated. (**D**) Analysis of how the *Dicer* knockout changes the amplitudes of rhythmic mRNA accumulation. Rhythmic transcripts from (**B**) are plotted according to how the ratio of the mRNA amplitude divided by the pre-mRNA amplitude is changed between *Dicer* knockout and control (abscissa; the further to the left a transcript is located, the lower the amplitude becomes in *Dicer* knockouts) and according to the change in mRNA/pre-mRNA ratio on the ordinate, as in (**C**). Red and black vertical lines correspond to mean amplitude changes of transcripts with and without increased mRNA/pre-mRNA ratios. The blue line shows the correlation of amplitude change over all transcripts (Pearson's r(670) = −0.0965; p=1.23e−02). Difference in amplitudes between the two groups of transcripts was tested using Wilcoxon rank sum test (W(672) = 36,518; p=9.39e−03). (**E**) Expression plots for six examples of transcripts whose phases are post-transcriptionally delayed in the absence of miRNAs. *Aco2, Fbxo21, Nampt, Ngfr, Pppde1* and *Sgpl1* thus all have similar phases of rhythmic transcription (pre-mRNA) in *Dicer* knockout (red) and control (blue) (right panels), but their mRNA accumulation is shifted to later times in the knockouts (left panels). Dotted lines represent the cosine curve fits. (**F**) Examples of transcripts showing amplitude effects. *Dnmt3a, Dnmt3b* and *Net1* thus all have similar transcriptional amplitudes in *Dicer* knockout (red) and control (blue) (right panels), but the loss of miRNAs reduces the mRNA amplitudes (left panels). (**G**) For rhythmically transcribed *Lipa*, the loss of miRNAs prevents rhythmic mRNA accumulation altogether. (**H**) *Cpeb1* mRNA shows the expected phase effect (later in *Dicer* knockout), but contrary to expectation, the mRNA amplitude is increased in the knockout.

The following source data and figure supplements are available for figure 5:

**Source data 1**.

**Figure supplement 1**. Analysis of phase and amplitude changes occurring upon *Dicer* knockout.

**Figure supplement 2**. Permutation of the RNA-seq data set confirmed that the effects observed on the phases and amplitudes (main *Figure 5C,D*) are highly significant and specific to the *Dicer* knockout.

A heatmap representation of the transcripts considered is shown in *Figure 5B* (see also *Figure 5—source data 1* for gene list).

We first determined how miRNA loss affected the phases of rhythmic mRNA accumulation (*Figure 5C*). We thus calculated the time lag between the transcripts' pre-mRNA and mRNA peaks for the *Dicer* knockouts, and subtracted the equivalent values calculated from the control mice. Positive and negative 'phase delay differences' thus signified that in the *Dicer* knockout, mRNA accumulation peaked later and earlier, respectively, than in controls (an additional analysis of absolute pre-mRNA and mRNA phases can be found in *Figure 5—figure supplement 1A,B*). Phase delay differences were plotted against the mRNA/pre-mRNA ratio change (*Figure 5C*; the 167 transcripts with a significantly higher ratio in the knockout are marked in red). This analysis revealed a shift to later phases of mRNA accumulation in the *Dicer* knockout, which correlated with increased mRNA/pre-mRNA ratios (blue line in *Figure 5C*; Pearson's r(670) = 0.17; p=9.24e−06). In the absence of miRNAs, mRNAs with significantly increased mRNA/pre-mRNA ratios thus peaked on average 48 min later (vertical red line in *Figure 5C*), whereas the average for all other rhythmic transcripts (vertical black line) lay at 5 min. Both groups of transcripts, that is those that showed higher mRNA/pre-mRNA ratios and those that did not, were significantly different (p-value 1.72e−07; Wilcoxon rank sum test; cumulative density plot at the top of *Figure 5C*). Although the globally detectable effect of miRNA loss on the phases of mRNA rhythms was relatively modest (less than 1 hr), individual transcripts were affected more dramatically (*Figure 5E*). For genes such as *Aconitase 2 (Aco2), F-box protein 21 (Fbxo21), Nicotinamide phosphoribosyltransferase (Nampt)* or *Nerve growth factor receptor (Ngfr)*, the phase-shifts amounted to several hours (*Figure 5E*). We also confirmed the observed effects for selected genes by qPCR (*Figure 4—figure supplement 5B*).

Using a similar approach, we next assessed how the loss of miRNAs influenced the amplitudes of mRNA rhythms. For each transcript, we calculated the ratio of the mRNA amplitude to the pre-mRNA amplitude in the *Dicer* knockout and divided this value by the corresponding ratio in control mice. We thus found that miRNA loss led to globally shallower amplitudes in the *Dicer* knockout (*Figure 5D*), although the mean effect was small and amounted only to a 7% decrease in fold-amplitude (see difference between red and black vertical lines in *Figure 5D*; p-value for difference between transcripts with and without increased mRNA/pre-mRNA ratio: 9.39e−03; Wilcoxon rank sum test). Nevertheless, individual examples clearly suggested that miRNAs assume important functions in ensuring that rhythmically transcribed genes give rise to rhythmic mRNAs that oscillate with the desired amplitudes and magnitudes, as shown in *Figure 5F* (qPCR validations in *Figure 4—figure supplement 5C,D*) for the

expression of *DNA methyltransferases 3A (Dnmt3a)*, *3B (Dnmt3b)* and *Neuroepithelial cell transforming gene (Net1)*, as well as for several other genes in *Figure 5E*. In some cases, such as *Lysosomal acid lipase A (Lipa)*, rhythmicity generated at the pre-mRNA level was completely lost at the mRNA level when miRNAs were absent (*Figure 5G*). Here, miRNA-mediated decay could thus represent the mechanism ensuring that after cyclic transcription, mRNAs are sufficiently unstable to show rhythmic accumulation at all. Interestingly, the protein encoded by *Lipa*, lysosomal acid lipase (LAL), is the key enzyme hydrolysing cholesteryl esters and triglycerides stored in lysosomes after LDL receptor-mediated endocytosis (*Fouchier and Defesche, 2013*) and has been reported to be rhythmic in liver, but non-rhythmic in other organs (*Tanaka et al., 1985*). Conceivably, miRNAs could be involved in rendering *Lipa* rhythms tissue-specific.

Altogether the effects of miRNA loss on amplitudes were less uniform than those on phases (compare *Figure 5C,D*). Several reasons could account for this difference. First, we noticed that amplitude estimations appeared technically more error-prone than phase estimations. Both parameters were read from cosine curves fitted to the data, but the peak-trough symmetry imposed by the cosine function frequently resulted in the underestimation of amplitudes; this was especially the case for high amplitude rhythms with pronounced 'spiky' appearance (i.e., with rapid rising and declining phases; see cosine fits of *Nr1d1/Rev-erbα* and *Dbp* in *Figure 3—figure supplement 1*). Second, miRNAs likely only represent one of several mechanisms operative in amplitude modulation; direct consequences of miRNA loss on target mRNA amplitudes might thus be partially masked by secondary effects occurring in the *Dicer* knockouts. Third, the model of how phases and amplitudes should ideally correlate (*Figure 5A*) is almost certainly an oversimplification. Indeed, for transcripts such as *Cpeb1* (*Figure 5H*), later phases were even accompanied by higher amplitudes. Globally, there was indeed no significant correlation between phase delays and amplitude decreases (*Figure 5—figure supplement 1E*). An uncoupling of amplitude and phase effects could occur, for example, through miRNA activity that is not constant over the day but confined to specific timepoints ('Discussion'). Overall, we concluded that the phase delays seemed to be the dominant, consistent consequence of miRNA loss in liver. Nevertheless, both the amplitude decrease and the phase delay were individually highly significant, as also shown by permutation tests in which we randomly reshuffled either the timepoints of the original data or the *Dicer* KO/control assignments on a genewise basis, followed by analyses of rhythm parameters. Among >10,000 datasets with permuted timepoints, the likelihood of finding similar or stronger delays in phase difference (in both size and significance) or correlations (between mRNA/pre-mRNA ratios and phase difference) than those in the original data were very low (*Figure 5—figure supplement 2A*). The observed shift in phase differences was therefore not an intrinsic data set property. By contrast, because the expression level differences (*Dicer* KO vs control) were untouched by the reshuffling of timepoints, a sizeable number of permutations still showed amplitude effects comparable to the original data (*Figure 5—figure supplement 2B*). However, when reshuffling occurred at the level of the genotypes, the permuted data no longer retained any trend reminiscent of the original data, neither for phases nor for amplitudes (*Figure 5—figure supplement 2C,D*). We concluded that the correlations uncovered between miRNA loss and rhythmic gene expression parameters were not likely to have arisen by chance, for example as a result of a specific predisposition of the data structure or distribution that could have occurred in our particular time series and the downstream analyses.

## A discrete group of miRNAs is predicted to specifically regulate circadian output pathways

The correct timing and extent of rhythmicity is important for the daily execution of clock-regulated physiological functions. It was thus likely that the observed phase and amplitude changes had an impact on liver functions. To identify pathways that were particularly affected by miRNA loss, we investigated whether the post-transcriptionally up-regulated circadian transcripts were associated with specific GO terms (*Figure 6A*). Using the same data set of circadian transcripts as before (*Figure 5*), several over-represented GO terms could be identified with statistical significance (FDR-adjusted p-value<0.05) in the group of rhythmic transcripts with increased mRNA/pre-mRNA ratios in *Dicer* knockouts, but not among the remaining rhythmic transcripts. However, the enrichment was relatively low, indicating that the function of miRNAs in regulating circadian gene expression was broad rather than confined to specific rhythmic functions. The individual GO terms affected three major fields: cell adhesion, apoptosis/development, and (lipid) metabolism. In particular, the latter caught our attention because serum analyses of *Dicer* knockout animals indicated metabolic phenotypes as well, especially

**A**

| GO term | (all genes in category) | circadian and mRNA/pre-mRNA ratio | | | |
|---|---|---|---|---|---|
| | | >0 (N=153) | | not >0 (N=469) | |
| | | genes | p-value (FDR adj.) | genes | p-value (FDR adj.) |
| Cell adhesion: Gap junctions | (30) | 4 | 9.98e-03 | 2 | 7.42e-01 |
| Cell adhesion: Endothelial cell contacts by junctional mechanisms | (26) | 3 | 4.67e-02 | 1 | 7.81e-01 |
| Cell adhesion: Tight junctions | (36) | 3 | 4.94e-02 | 2 | 7.59e-01 |
| Putative pathways for stimulation of fat cell differentiation by Bisphenol A | (32) | 3 | 4.94e-02 | 3 | 5.48e-01 |
| Regulation of metabolism: Bile acids regulation of glucose and lipid metabolism via FXR | (37) | 3 | 4.94e-02 | 2 | 7.72e-01 |
| Apoptosis and survival: Regulation of Apoptosis by Mitochondrial Proteins | (33) | 3 | 4.94e-02 | 1 | 7.81e-01 |
| Development: MAG-dependent inhibition of neurite outgrowth | (37) | 3 | 4.94e-02 | 0 | 1.00e+00 |

**B**

| miRNA seed family* | circadian and mRNA/pre-mRNA ratio > 0 (N=167) | | | |
|---|---|---|---|---|
| | expected # targets | actual # targets | enrichment (MLE)** | p-value (FDR adjusted) |
| miR-320abcd/4429 | 35.5 | 57 | 1.922 | 6.97e-03 |
| miR-361 | 20.8 | 39 | 2.144 | 6.97e-03 |
| miR-25/32/92abc/363/367 | 19.3 | 36 | 2.109 | 9.01e-03 |
| miR-122/1352 | 22.1 | 39 | 1.993 | 1.16e-02 |
| miR-874 | 26.4 | 44 | 1.904 | 1.16e-02 |
| miR-22 | 27.8 | 45 | 1.846 | 1.34e-02 |
| miR-29abcd | 26.3 | 43 | 1.855 | 1.34e-02 |
| miR-326/330 | 39.1 | 58 | 1.738 | 1.34e-02 |
| miR-377 | 37.7 | 55 | 1.683 | 2.58e-02 |

\* red: good expression; black: detectable, but low expression; grey: not detectable by small RNA-seq.
\*\* conditional Maximum Likelihood Estimate (MLE) of the odds ratio.

**C**

**D**

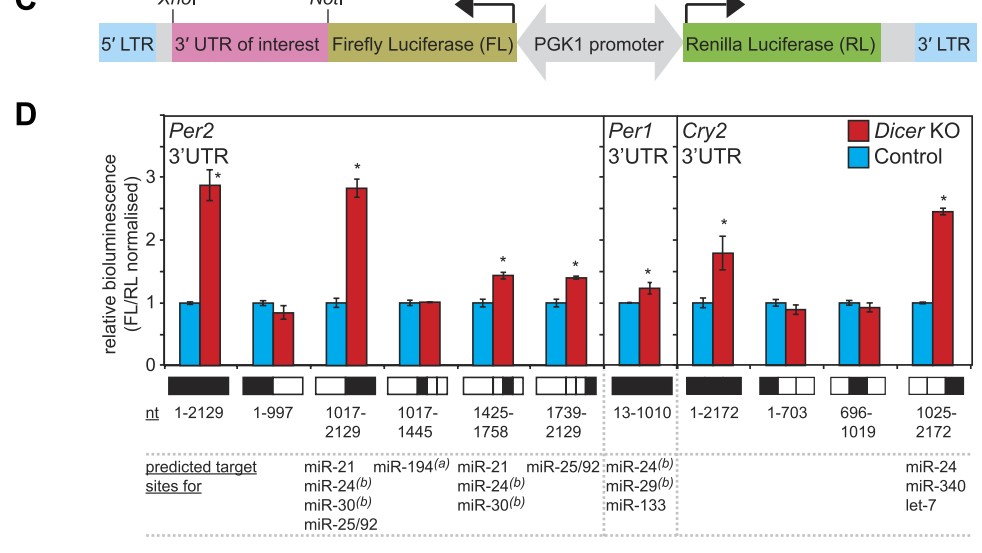

Figure 6. Regulation of circadian output pathways by specific miRNAs. (**A**) GO term analysis identifies specific pathways enriched in the group of circadian transcripts that are likely miRNA-regulated (higher mRNA/pre-mRNA ratio in *Dicer* knockouts) but not so in the remainder of rhythmic mRNAs; p-values are corrected for FDR due to multiple testing. Deviations from transcript numbers in *Figure 5* are due to genes without associated GO term. (**B**) List of miRNAs for which more predicted targets than expected are found in the group of circadian transcripts with higher mRNA/pre-mRNA ratios in *Dicer* knockout. The miRNAs marked in red are expressed at readily detectable levels in liver; those in black were identified in liver by small RNA-seq, but only at very low levels. For miRNAs in grey, we found no evidence for expression in liver by RNA-seq. miRNA predictions performed with targetscan ('Materials and methods'). (**C**) Reporter construct to study the function of 3′ UTRs sequences. Within a lentiviral expression cassette (defined by the long terminal repeats, LTRs, in blue), two different luciferase mRNAs are transcribed from the bidirectional *Pgk1* promoter. Firefly luciferase serves as the reporter gene to test the effect of a particular 3′ UTRs (pink), whereas renilla luciferase serves as a control for internal normalisation. (**D**) Effect of *Per2*, *Per1* and *Cry2* 3′ UTRs on reporter expression in control (blue) and *Dicer* knockout (red) primary hepatocytes. Lentivirally delivered full-length *Per2* 3′ UTRs-containing reporter (nt 1–2129) is thus de-repressed in *Dicer* knockouts; this effect is mediated by the second half of the UTR (1017–2129), probably

*Figure 6. Continued on next page*

*Figure 6. Continued*

in synergy through fragment 1425–1758, which contains the predicted miR-24 and miR-30 sites from *Chen et al. (2013)*, and fragment 1739–2129, which contains a predicted site for the miR-25/92 family. *Per1* UTR is only slightly de-repressed in *Dicer* knockout hepatocytes. *Cry2* appears to be miRNA-regulated by sites located in the 3'-terminal portion of the UTR, which contains predicted sites for miR-24, miR-340 and let-7. The miRNAs listed below the graph represent targetscan predictions filtered for those detected in liver by small RNA-seq. (**A**) and (**B**) identify miRNA regulation reported by *Nagel et al. (2009)* and *Chen et al. (2013)*, respectively. Data correspond to mean ± standard deviation from triplicate assays from independent lentiviral transductions using hepatocytes from the same mice. Each experiment was confirmed at least twice using hepatocytes from independent animals.

The following figure supplements are available for figure 6:

**Figure supplement 1**. Serum analyses in *Dicer* knockouts indicate metabolic defects.

for cholesterol, triglyceride, and glucose levels, which were all reduced (*Figure 6—figure supplement 1*). At least in part, these effects were likely mediated by the loss of miR-122, whose inactivation has previously been shown to impact lipid metabolism (*Krutzfeldt et al., 2005*; *Esau et al., 2006*; *Gatfield et al., 2009*; *Wen and Friedman, 2012*).

The breadth of miRNA loss (including potential secondary effects) poses an obvious limit to the suitability of *Dicer* knockouts for further functional studies. Ideally, the targeted inactivation of single miRNAs that fulfil prospective functions in circadian regulation would allow for more straightforward investigations. To identify such miRNAs, we analysed whether the circadian transcripts that became more stable in *Dicer* knockouts were enriched for predicted miRNA binding sites (*Lewis et al., 2005*). Interestingly, this analysis revealed nine miRNA seed families with more circadian targets than expected by chance (*Figure 6B*) specifically in the data set with higher mRNA/pre-mRNA ratios. Of these, miR-25, miR-92, miR-122, miR-22, and miR-29 are expressed at readily detectable levels in liver, as shown previously (*Landgraf et al., 2007*; *Gatfield et al., 2009*; *Vollmers et al., 2009*) and confirmed by northern blot (*Figure 1—figure supplement 2*) and small RNA-seq (data not shown). Given that miR-122 already has a history as a circadian output modulator (*Gatfield et al., 2009*), it is tempting to speculate that the other miRNAs may represent specialised regulators of the rhythmic transcriptome as well. Interestingly, miR-29 is also one of the miRNAs that was reported to regulate *Per1* in MEFs (*Chen et al., 2013*). In the future, it would thus be exciting to explore the role of these miRNAs in targeted experiment in the context of clock-regulated biological pathways.

## Validation of 3' UTRs using a reporter assay in primary hepatocytes predicts *Per2*-regulating miRNAs

The combination of increased mRNA/pre-mRNA ratios and miRNA target predictions is suggestive of bona fide miRNA-mediated regulation. Nevertheless, before embarking on follow-up experiments for individual transcripts, validation experiments to map miRNA binding sites in the targets' 3' UTRs and to identify the responsible miRNAs will be necessary. To this end, we designed a luciferase reporter assay based on the lentiviral transduction of *Dicer* knockout and control primary hepatocytes (*Figure 6C*). We chose the core clock transcripts to demonstrate the utility of the assay, as we were particularly intrigued by the difference in phenotypes between the *Dicer* knockout in liver (i.e., mild period length-ening, *Figure 3E*) and the previously reported strong period shortening in MEFs (≈2 hr), which the authors attributed to miRNA-regulation of *Per1* (by miR-24 and miR-29) and *Per2* (by miR-24 and miR-30) (*Chen et al., 2013*). We tested whether the firefly luciferase reporter gene carrying the full-length *Per1*, *Per2* or *Cry2* 3' UTRs (or fragments thereof) was subject to de-repression in *Dicer* knockout hepatocytes, as compared to control hepatocytes. Consistent with the RNA-seq and western blot analyses (*Figure 3B,C*), this assay indeed identified *Per2* as the main miRNA-regulated core clock component (*Figure 6D*). Moreover, truncated versions of the 3' UTR that contained the miR-24 and miR-30 sites identified by Chen et al. recapitulated some of the miRNA regulation observed for the full-length *Per2* 3' UTR sequence. However, additional regulation was also conferred by a portion of the 3' UTR containing a predicted target site for the miR-25/92 seed family (fragment 1739–2129, *Figure 6D*) and full regulation may thus require the synergistic activity of several sites. *Per1* and *Cry2* full-length 3' UTRs showed 1.2-fold and 1.8-fold up-regulation, respectively (*Figure 6D*). This was in the same range as what we had observed for the endogenous proteins by western blot (*Figure 3C*). In particular, the lack of strong *Per1* regulation in liver was a striking difference to the MEF data and could be responsible for phenotypic differences, as Chen et al. have proposed that the faster translation

and higher accumulation of PERs (approximately twofold higher PER1 and PER2 peak levels in *Dicer* knockout) shortens the time delay within the oscillator's main feedback loop and speeds up the clock. It is possible that tissue-specific miRNA expression contributes to the observed differences; in line with a previous study (*Vollmers et al., 2012*), we have thus observed that miR-30 is among the most abundant miRNA species in liver (within the top 10 of miRNAs detected by small RNA-seq), but that miR-29 and miR-24 are expressed at lower levels (not in the top 50) (data not shown), which could thus partially explain the comparably weak effect of the *Dicer* knockout on PER1 in liver. In summary, in addition to shedding light on miRNA regulation of core clock transcripts, we concluded that the reporter assay in hepatocytes represented a useful tool for future studies aimed at verifying direct miRNA targets and at unravelling the contribution of specific miRNAs.

## Discussion

Recent publications have suggested widespread contributions of post-transcriptional regulation to circadian mRNA cycling and have challenged the assumption that rhythmic transcription is the main driver of rhythmic gene expression (*Koike et al., 2012*; *Le Martelot et al., 2012*; *Menet et al., 2012*). MicroRNAs are important post-transcriptional regulators (*Bushati and Cohen, 2007*; *Krol et al., 2010*; *Fabian and Sonenberg, 2012*) whose functions in mammalian circadian biology are only beginning to emerge (*Lim and Allada, 2013*; *Mehta and Cheng, 2013*). We have used a drastic approach, that is the knockout of the miRNA biogenesis factor *Dicer* (*Harfe et al., 2005*), to assess how the rhythmic transcriptome in mouse liver is altered in the absence of miRNA-mediated regulation (*Figure 7*). Considering the severity of the genetic model, our finding that the circadian system was globally very resilient to miRNA loss was surprising and somewhat reminiscent of the stability of circadian oscillators with regard to other perturbations such as large fluctuations in general transcription rates and temperature (*Dibner et al., 2009*). Moreover, our findings are in line with the notion that miRNAs frequently function in the fine-tuning and modulation of gene expression (*Krol et al., 2010*; *Fabian and Sonenberg, 2012*; *Yates et al., 2013*).

The hepatic core clock remained fully functional and showed only modest period lengthening of free-running rhythms (on average by 40 min) in *Dicer* knockout liver explants. This phenotype is a plausible consequence of the approximately twofold increase in expression of PER2, which we identified as the main miRNA-regulated core clock component in hepatocytes. Mice in which *Per2* levels are increased due to additional *Per2* transgenes thus show a similar period lengthening (*Gu et al., 2012*), while *Per2* knockout mice have short periods before becoming arrhythmic (*Zheng et al., 1999*; *Bae et al., 2001*). Conceivably, the longer period seen in *Dicer* knockout liver explants is masked in the animal due to constant entrainment by the SCN and other cues. It is well possible that a phenotype will manifest in knockouts only under conditions that bring the steady-state relationship between entrainment cues and the liver clock out of equilibrium, such as in jet lag or food shifting experiments (*Damiola et al., 2000*); this situation would be consistent with models stating that miRNAs confer robustness to oscillatory networks and denoise negative feedback loops (*Cohen et al., 2006*; *Gerard and Novak, 2013*). Intriguingly, *Per2* has already been noted to act as a link between systemic entrainment signals and local liver clocks (*Kornmann et al., 2007*). It would thus be exciting to investigate clock readjustment kinetics in *Dicer* knockouts, for example using a novel method for the real-time recording of liver rhythms (*Saini et al., 2013*).

Interestingly, the previously reported dramatic period shorting (≈2 hr) in *Dicer* knockout MEFs (*Chen et al., 2013*) is in stark contrast to what we observed in liver. Very likely, tissue-specific differences in miRNA activity form the basis of these differences and could explain that PER1 and PER2 are equally strongly affected in *Dicer* knockout MEFs, whereas in liver PER1 appears to be less strongly regulated. Interestingly, an earlier study on the SCN clock has revealed brain-specific miR-219 and miR-132 as regulators of period length and light-induced clock resetting, respectively (*Cheng et al., 2007*). It is thus tempting to speculate that miRNAs generally function to post-transcriptionally tune the core clock in a cell type-specific fashion. At least in part, miRNA activity could thus also underlie known tissue-specific differences in phases and free-running period lengths (*Yoo et al., 2004*).

The similarity of core clocks proved advantageous for the comparison of clock-controlled gene expression between *Dicer* knockouts and controls animals. To this end, we found that miRNAs contributed only marginally to generating transcript rhythms at the post-transcriptional level. Less than 2% of all rhythmic mRNAs thus fulfilled our criteria for miRNA-driven rhythmicity and the identified examples (e.g., *Figure 4E–I*) showed low amplitude cycling. This finding is consistent with the notion that miRNAs

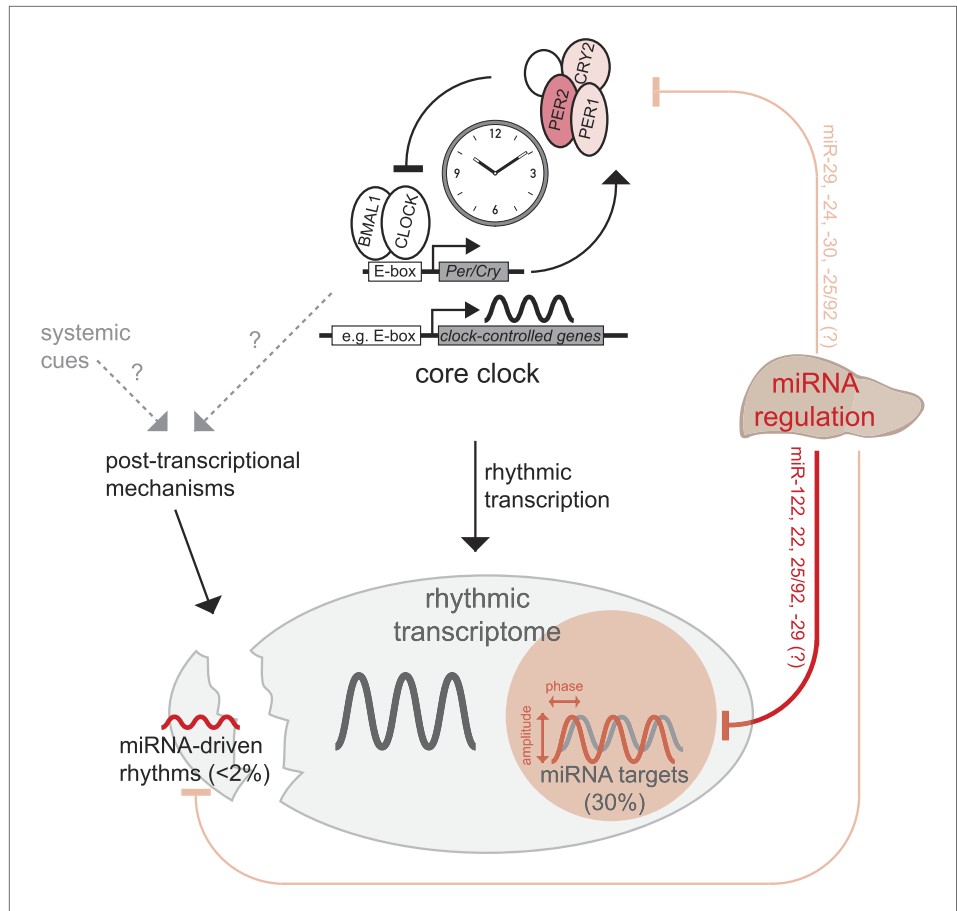

**Figure 7**. Model, summary, and speculations. In the liver, we propose that miRNAs play three distinct roles in the regulation of rhythmic gene expression. First, around 30% of rhythmically transcribed genes appear to be also regulated by miRNAs, which tunes the phases and amplitudes of mRNA accumulation. This group of transcripts is enriched for predicted binding sites for several miRNAs (**Figure 6B**), such as miR-122, miR-22, miR-25/92, and others. Second, for a very small group of transcripts (<2% of all rhythmic mRNAs) rhythms may be driven by miRNAs, but miRNA activity is unlikely to underlie major discrepancies between the rhythmic transcriptome and rhythmic transcription. Finally, miRNA activity seems to be dispensable for a functional hepatic core clock. Nevertheless, it is conceivable that under conditions where the clock is brought out of equilibrium and has to readjust (e.g., jet lag, food shifting) the identified miRNA-mediated regulation in particular of *Per2* (but also of *Per1* and *Cry2*) would be of functional importance.

are usually highly stable molecules and generally not expected to show pronounced daily variations in abundance (see below). By contrast, we have found that miRNAs exert important functions in refining the rhythmic mRNA accumulation profiles of cyclically transcribed genes. We thus estimate that for about 30% of the rhythmic transcriptome, miRNA-mediated regulation adds an additional layer of post-transcriptional control. Concretely, the absence of miRNAs caused a global shift of mRNA accumulation to later phases during the day. Moreover, peak-to-trough amplitudes of mRNA accumulation were overall reduced in *Dicer* knockouts, although this effect was globally smaller and more variable across the transcriptome. These results establish that miRNAs are in charge of an important regulatory control level sandwiched between rhythmic transcription and the final rhythmic mRNA and, eventually, protein output. Because many miRNAs are expressed in a tissue-specific fashion, they may thus be key to converting rhythmic transcriptional information into the desired tissue-specific rhythmic outcome. Our miRNA target prediction analysis would suggest that a discrete group of miRNAs (miR-25/92, miR-122, miR-22 and miR-29) is specifically involved in the post-transcriptional tuning of circadian transcripts in the liver. Using the lentiviral reporter system, it will be possible to confirm some of the predicted miRNA-target interactions in hepatocytes, which could then form the basis for targeted experiments

in which the expression of individual miRNAs is manipulated, for example by genetic or antisense loss-of-function techniques.

Constantly expressed miRNAs can explain many of the observed phase and amplitude effects (*Figure 5A*). For transcripts such as *Lipa*, miRNA-mediated regulation may thus merely represent a convenient mechanism to keep mRNAs sufficiently unstable to ensure that they cycle at all. However, the post-transcriptional profiles of transcripts such as *Ddx17*, *Slc1a15* (*Figure 4E,F*), or *Cpeb1* (*Figure 5H*), could be suggestive of rhythmic miRNA activity. Several miRNAs have been previously reported as potentially rhythmic in mouse liver (*Na et al., 2009*; *Vollmers et al., 2012*), but our small RNA-seq profiling (data not shown), as well as previous analyses of individual miRNAs (*Gatfield et al., 2009*) could not confirm high-cycling miRNA species. Similarly, the recently reported circadian expression of *Dicer* itself (*Yan et al., 2013*) was not evident from our data (*Figure 2—figure supplement 2B*). Collectively, these findings suggest that the potential for rhythmic miRNA activity in mouse liver is relatively low. However, our approach, which analyses miRNA activity almost exclusively from the perspective of miRNA targets, cannot resolve this issue; a dedicated study would thus be informative.

The *Dicer* knockout allele that we have used (*Harfe et al., 2005*) has previously served as an entry point to define the role of miRNAs in many other fields (e.g., *Harris et al., 2006*; *Chong et al., 2008*; *Sheehy et al., 2010*). In spite of the insinuated universality of miRNA loss, there are some limitations to the approach. First, in a few exceptional cases, miRNA biogenesis can be independent of canonical Dicer processing, for example for miR-451 (*Yang and Lai, 2011*). This miRNA is indeed not depleted from *Dicer* knockout livers (*Figure 1—figure supplement 2B* and data not shown). Moreover, similar to most other studies, we deduced miRNA activity from target mRNA abundance. This approach seems justified overall because genome-wide measurements of miRNA action on mRNA and protein levels have shown that these correlate generally well (*Baek et al., 2008*; *Selbach et al., 2008*). Nevertheless, it should be kept in mind that miRNAs may affect mRNA levels less strongly than protein levels (*Selbach et al., 2008*; *Yang et al., 2010*), and that mRNA changes can even be absent altogether (e.g., *Bhattacharyya et al., 2006*). Finally, distinguishing direct from indirect effects in *Dicer* knockout data is challenging. We have used a measure of mRNA stability (mRNA/pre-mRNA ratios) to deplete our dataset of gene expression changes that involve altered transcription and that are probably mostly indirect. However, in specific cases miRNAs have been reported to directly interfere with transcription through promoter complementarity (reviewed in *Huang and Li, 2012*). Obviously, our analyses miss direct effects of miRNAs that are not post-transcriptional. On the other hand, the use of increased mRNA stability (mRNA/pre-mRNA ratios) to enrich for likely direct miRNA targets will inevitably result in false-positives that are post-transcriptionally regulated due to indirect effects, for example because components of the mRNA decay machinery are direct miRNA targets. In spite of these caveats, our study represents a comparatively complete analysis of miRNA activity in liver and should prove a valuable resource for further investigations of circadian and non-circadian functions that these regulatory molecules exert in hepatic gene expression, metabolism, and physiology.

## Materials and methods

### Animal care and treatment

Animal studies were conducted in accordance with the regulations of the veterinary office of the Canton of Vaud (authorization VD2376). All alleles used in the study have been published before that is, *Dicer^flox* (*Harfe et al., 2005*), *Alb^Cre-ERT2* (*Schuler et al., 2004*), and *mPer2^Luc* (*Yoo et al., 2004*), and were kindly provided by the Tabin, Metzger, and Takahashi labs, respectively. The genetic background of the animals used in this study was mixed. Male littermate knockout (*Dicer^flox/flox*;*Alb^Cre-ERT2*) and control (*Dicer^flox/+* or *Dicer^+/+*;*Alb^Cre-ERT2*) animals aged 3–6 months received tamoxifen treatment by intraperitoneal injections over 5 days (total 2 mg) essentially as described (*Schuler et al., 2004*). We combined both wild-type and heterozygous male littermates for the controls because microarray analyses indicated that a single functional *Dicer* allele was sufficient to ensure miRNA processing to wild-type levels (*Figure 1—figure supplement 2A*), as expected from previous studies (*Harfe et al., 2005*; *Kanellopoulou et al., 2005*; *Murchison et al., 2005*; *Chen et al., 2008*; *Frezzetti et al., 2011*). Haploinsufficiency has indeed so far only been reported for tumour suppressor functions of *Dicer* in certain non-liver tumorigenesis models (*Kumar et al., 2009*; *Arrate et al., 2010*; *Lambertz et al., 2010*; *Nittner et al., 2012*; *Yoshikawa et al., 2013*).

After the last injection, mice were entrained to a 12-hr light:12-hr dark photoperiod with free access to food and water for 1 month, sacrificed at the respective Zeitgeber times (ZT 0, 4, 8, 12, 16, 20), and the liver was snap-frozen in liquid nitrogen. Oligos used to genotype *Dicer* knockout efficiency from genomic DNA (*Figure 1—figure supplement 1A*) were DicerF1, DicerR1, and DicerDel (see Supplementary file 2 in *Du et al., 2014* for sequences).

## RNA preparation and analysis

Total RNA was prepared essentially as described previously (*Gatfield et al., 2009*). The efficiency of the *Dicer* knockout was confirmed for each liver sample by northern blot probing for miR-122. The protocol for miRNA northern blots has been described (*Gatfield et al., 2009*). Total RNA pools for the two time series in knockouts and controls were assembled using identical amounts of RNA from 3 to 4 animals per pool. All pools were DNase-digested (RQ1 DNase, Promega, Madison, WI) in order to eliminate potential contamination from genomic DNA that would have rendered pre-mRNA quantifications impossible. The absence of genomic DNA contamination was confirmed in all pools by the comparison of RT− and RT+ reactions in quantitative real-time PCR (qPCR) using genomic probes. For qPCR analysis, cDNA was synthesised from 6 μg of DNase-treated total RNA using random hexamers and SuperScript II reverse transcriptase (Invitrogen, Carlsbad, CA) according to the supplier's instructions. cDNA was PCR-amplified using FastStart Universal SYBR Green Master (Roche, Basel, Switzerland). Mean levels were calculated from triplicate PCR assays for each sample and normalised to those obtained for the control transcripts *GusB* and *Eef1a1* (*Figure 1E*) and *Csnk1a1, Ctsd, Nudt4, Smg1* and *Trip12* (*Figure 4—figure supplement 5*). All oligonucleotide sequences are listed in Supplementary file 2 (*Du et al., 2014*).

## miRNA microarrays

Total RNAs (100 ng; same samples as for the RNA-seq series) were hybridised to Mouse miRNA Microarray, Release 18.0, 8 × 60K (Agilent Technologies, Santa Clara, CA) using miRNA Complete Labelling and Hyb Kit (Agilent Technologies) according to the supplier's instructions. An invariant normalisation method using five internal spikes (hur_1, hur_2, hur_4, hur_6, mr_1) was used to normalise *Dicer* KO vs control samples in *Figure 1—figure supplement 2B*. Quantile normalisation was used for heterozygote knockout vs wild-type samples in *Figure 1—figure supplement 2A*.

## Plasmids, cloning of 3′ UTRs, lentiviral production

The dual luciferase reporter cassette in which the bidirectional *phosphoglycerate kinase 1* (PGK1) promoter (*Amendola et al., 2005*) drives the expression of firefly luciferase (FL, used for 3′ UTR cloning) and renilla luciferase (RL, for normalisation) was designed *in silico* and purchased as a synthetic clone (GenScript, Piscataway, NJ) in pUC57 plasmid. After excision from the plasmid via flanking *Sal*I sites, the cassette was used to replace the *Xho*I cassette in lentiviral plasmid pWPT-GFP (Addgene, Cambridge, MA), resulting in plasmid prLV1 that was used to clone 3′ UTRs of interest via *Xho*I/*Not*I restriction sites downstream of the firefly luciferase coding sequence. 3′ UTR sequences were amplified by PCR from liver cDNA or genomic DNA with specific oligonucleotides that carry *Not*I sites in the forward (F) and *Xho*I sites in the reverse (R) oligos, as listed in Supplementary file 2 (*Du et al., 2014*). The identity of the cloned UTRs was verified by sequencing.

From the prLV1 vectors with cloned UTRs, lentiviral particles were produced in 293T cells using envelope vector pMD2.G and packaging plasmid psPAX2 as previously described (*Salmon and Trono, 2007*). Viral supernatant was spun 2 hr at 25,000 rpm, 4°C using Optima L-90K Ultracentrifuge (SW32Ti rotor; Beckman, Brea, CA), then viral particles were resuspended with primary hepatocyte medium.

## Primary hepatocytes preparation and viral transduction

Mice were anesthetised and livers were perfused through the inferior vena cava with 50 ml of washing buffer (137 mM NaCl, 2.7 mM KCl, 0.5 mM Na$_2$HPO$_4$, 10 mM HEPES, pH 7.65) supplemented with 0.5 mM EDTA, then 50 ml of digestion buffer (washing buffer supplemented with 7 mM CaCl$_2$, 0.4 mg/ml collagenase [C5138; Sigma-Aldrich, St. Louis, MO]) at flow rate of 5 ml/min and at 37°C. Isolated cells were filtered with a cell strainer (100 μm, BD Falcon, Franklin Lakes, NJ) and washed with 40 ml of primary hepatocyte medium (Medium 199, GlutaMAX Supplement, supplemented with 1% penicillin/streptomycin/glutamine (PSG), 0.1% BSA, 10% FCS [all Gibco/Life Technologies, Carlsbad, CA]). The cells were resuspended with 10 ml of primary hepatocyte medium, counted, and plated in 12-well plates (coated with 0.2% gelatin) at a density of 2 × 10$^5$ cells/well. Medium was changed 4 hr later and viral supernatant was added. The cells were maintained at 37°C and

5% $CO_2$ until harvested. For each lentiviral construct, transductions were performed in triplicates. Each experiment was repeated at least twice with cell preparations from independent mice.

## Luciferase assay

6 days after lentiviral transduction, cells were collected using 5x Passive Lysis Buffer (Promega) and luciferase activity was measured using the Dual-Glo Luciferase Assay System (Promega) according to the manufacturer's protocol. Firefly luciferase signals were normalised to Renilla luciferase, and for each 3′ UTR construct this signal was then normalised to that of lentivector-control plasmid (without cloned 3′ UTR, thus only containing generic vector 3′ UTR). Signals in control mice were set to 1.

## Bioluminescence analysis of liver explants

Mice were anesthetised with isoflurane and sacrificed by decapitation. The liver was excised and immediately placed in cold Hank's balanced salt solution (Invitrogen). Then, the liver was sliced into small pieces and cultured separately on Millicell culture membranes (PICMORG50; Millipore, Billerica, MA) with 1.1 ml of HEPES-buffered phenol red-free DMEM (Gibco) supplemented with 2% B27 (Invitrogen), 1% PSG (Gibco), 4.2 mM $Na_2HCO_3$ and 0.1 mM luciferin. Cultures were maintained at 37°C and 5% $CO_2$ in a light-tight incubator, and bioluminescence was monitored continuously using LumiCycle 32 (Actimetrics, Wilmette, IL).

## Protein extraction and immunoblot analysis

Proteins from mouse liver nuclei were prepared according to the NUN procedure (*Lavery and Schibler, 1993*). Each sample analysed by SDS-PAGE was a pool of protein extracts from 3 to 5 mice. SDS-PAGE and immunoblot analysis were performed according to standard protocols. Antibodies used were rabbit CRY1, CRY2, PER1, PER2, BMAL1, and CLOCK (kindly provided by S Brown and J Ripperger) and U2AF65 (Sigma). Western blots were quantified using ImageQuant TL 8.1 software (Life Sciences).

## Blood chemistry

All blood chemistry was performed as described (*Le Martelot et al., 2009*).

## cDNA library preparation and sequencing by RNA-seq

DNase-treated total RNAs were subjected to rRNA depletion using Ribo-Zero Magnetic Kit (Human/Mouse/Rat, Epicentre, Madison, WI) following the supplier's instructions. The resulting rRNA-depleted RNA samples were then used to prepare random-primed cDNA libraries using the Illumina Tru-Seq RNA Sample Preparation Kit (Illumina, San Diego, CA) according to the manufacturer's recommendations. Multiplexed libraries were sequenced following the supplier's protocol (100 bp single-end reads) on the Illumina HiSeq 2500 at the Lausanne Genomics Technologies Facility.

## RNA-seq: quality assessment of sequencing

Quality of the sequencing reads was initially assessed based on the quality values produced by the Illumina pipeline Casava 1.8.2. Sequencing runs were only then allowed into our data analysis work flow, if their statistics related to quality of base calling and to preliminary alignment against mouse genome were within three standard deviations of the mean of all runs. The five quality related statistics used were the percentage of clusters passed filtering (%PF clusters), mean quality score (PF clusters), percentage of reads aligned, mean alignment score and percentage of alignment error, and had the following population means and standard deviations, respectively, 93.39 ± 1.85, 36.34 ± 0.5, 72.24 ± 3.0, 290.6 ± 15.13 and 0.98 ± 0.15. Presence of adapter sequences in the datasets was checked with cutadapt utility (*Martin, 2011*). Less than 0.65% of the reads were estimated to include an adapter sequence longer than 10 bps; and more than 95% of such reads were left unmapped after the alignment of the reads to the mouse genome. Therefore, it was not necessary to trim reads to remove the adapter sequences.

## Alignment of RNA-seq data to the mouse genome

Sequences were first mapped to local databases of mouse rRNA (Ensembl and NCBI) and rodent-specific repeat sequences (RepBase Update version 18.10 [*Jurka et al., 2005*]) using bowtie2 version 2.1.0 (*Langmead and Salzberg, 2012*) with default alignment parameters. Then reads were mapped to Genome Reference Consortium GRCm38 (mm10) version of mouse reference genome sequence using tophat version 1.4.1 with known mm10 transcripts provided via the—transcriptome-index option (*Trapnell et al., 2009*). For each read all mapping outcomes were considered and only non-rRNA,

non-repeat reads that mapped uniquely to the mouse genome were selected for further analysis (*Figure 2—source data 1*).

## Quantification of mRNA and pre-mRNA expression levels from RNA-seq

Expression levels for mRNA and pre-mRNA were estimated per locus rather then per isoform. To this end, gene annotations from Ensembl mouse database release 68 (*Flicek et al., 2013*) were flattened similarly as described earlier (*Anders et al., 2012*) with the following modifications. Prior to flattening, transcript isoform models whose expression was not evidenced by count data or splice junction coverage were removed from the annotation database. This was achieved first by estimating the expression level of each isoform contained in Ensembl database by cufflinks software version 2.0.2 (*Trapnell et al., 2010*) with following parameters: -G Mus_musculus.GRCm38.68.gtf -u -j 0.8 -m 160 -s 50 -N. Isoform models which did not comprise at least 5% of the total expression of the locus or which were only expressed in fewer than three samples were excluded from the gene annotation database. Furthermore, spliced isoform models, whose splice junctions were not covered on average by 6 reads, or 40% or more of whose splice junctions were not covered at least by 6 reads were also removed from the annotation database. In addition to the removal of non-supported isoform models, novel exons overlapping with reference on the opposite strand were identified and added to the annotation database to improve the unambiguity of read counting. Briefly, cufflinks was run in reference annotation based transcript (RABT) mode (*Roberts et al., 2011*) and isoform models were assembled with cuff-merge software. Novel exon models with a class code 'X', which overlap with a reference exon on the opposite strand, were then validated by splice junction coverage as outlined before and added to the annotation database. Finally, the gene models were flattened (*Anders et al., 2012*).

An in-house Python script was used to count the reads mapped within each annotation feature in a similar way as implemented in htseq-count utility software (HTSeq: analysing high-throughput sequencing data with Python. [http://www-huber.embl.de/users/anders/HTSeq/]). Only reads which can be unambiguously identified as either exonic (continuous or spliced) or intronic for a single locus were counted towards the mRNA or pre-mRNA counts of that locus, respectively. Mappable and countable mRNA and pre-mRNA lengths (in bps) for each locus were calculated by means of generating all possible 100-bp long reads in silico (faux reads) for each transcript type and counting the faux reads through identical mapping and counting work flow used for real experimental reads.

Preliminary inspection of the extent of differential expression and the presence of highly expressed genes was carried out by cumulative percentage plots of raw counts; and accordingly read counts of mRNA and pre-mRNA datasets were normalised with upper quantile (*Bullard et al., 2010*; *Dillies et al., 2013*) and TMM (*Robinson and Oshlack, 2010*) normalisation methods, respectively. Prior to normalisation, transcripts which did not have at least 10 counts in at least one third of the samples were removed from the datasets. Differential expression (DE) analysis was performed using the DESeq analysis tool version 1.12.1 (*Anders and Huber, 2010*). Briefly, normalisation factors calculated earlier were used to create DESeq count datasets for mRNA and pre-mRNA counts annotated with full experimental design (treatment × time) and dispersion values were calculated with default settings. Calculated p-values for DE of all transcripts between control and *Dicer* KO conditions were then adjusted for false discovery rate (FDR) by Benjamini and Hochberg (BH) method (*Benjamini and Hochberg, 1995*). A transcript was considered differentially expressed if it had an adjusted p-value smaller than 0.05 and had a fold-change greater than 1.5.

For further filtering of lowly expressed transcripts and for better comparability between datasets, RPKM values were calculated as the number of counted reads per 1000 mappable and countable bases per geometric mean of normalised read counts per million. The geometric mean of normalised read counts was 53,692,658 and 16,285,588 for mRNA and pre-mRNA datasets, respectively. Transcripts which did not have a RPKM value greater than 0.1 or 0.01 in at least one third of the samples were considered as 'not expressed' in the mRNA or the pre-mRNA dataset, respectively.

As an estimate of the mRNA stability, we have calculated the log2 mRNA to pre-mRNA ratios of RPKM values separately for each sample. The distribution of log-ratios was inspected to assess normality. To test if a transcript's mean stability over all timepoints was increased in the absence of miRNAs, we applied a one-sided *t* test between control and KO conditions (n = 12). The p-values were then adjusted for FDR using BH method.

## Rhythmicity detection in transcript profiles

We used the combined power of a parametric test based on harmonic regression similar to those implemented in cosiner based algorithms (*Cugini, 1993*; *Yang and Su, 2010*) and a non-parametric test, JTK_CYCLE (*Hughes et al., 2010*). To adapt the harmonic regression to sequencing count data, several modifications were introduced. First, a negative binomial generalised linear model (GLM), with a log link function was used to regress the rounded normalised counts on a cycle with a 24-hr period. Count data were not detrended or smoothened before the regression. Dispersion values that were already fitted for each gene during DE analysis were used as an estimate of the shape parameter theta. From the fitted parameter values, then, amplitude and phase estimates were calculated for each gene. The goodness-of-fit for the harmonic model (sinusoidal expression) was tested with a likelihood ratio test against the updated null model including only the intercept parameter (constant expression). The analysis was performed using the MASS package (*Rindskopf, 1997*) in statistical computing environment R.

For both tests, the two biological replicates per timepoint were treated as independent replicates and were not combined into a false 48-hr data-series. The p-values calculated by each test were then adjusted separately for FDR by BH method (*Benjamini and Hochberg, 1995*). For all further calculations, amplitude and phase estimates obtained from the harmonic fit were used. We categorised mRNAs and pre-mRNAs as rhythmic when they passed at least one of the statistical tests and had an amplitude-ratio (peak/trough) ≥1.5. Combining two complementary tests in this way provided for relatively low stringency in rhythm detection, which we deemed important in particular for the pre-mRNA data, which is naturally noisier due to the low abundance of short-lived pre-mRNA molecules.

## Permutation tests

To assess the significance of the effects of the treatment on amplitude and phase estimates, we performed two permutation tests (N > 10,000). For both tests, we applied the permutation on the normalised read counts per gene, always maintaining the mRNA and pre-mRNA attributes. Each permuted data set was then analysed by exactly the same steps used on the experimental data. Finally, the distribution of the difference in means, significance of the effects, and correlation coefficients under the null hypothesis were inspected for both amplitude and phase effects. To enable the comparison of the significance of the effects between samples of different sizes, the Wilcoxon test statistics were converted into a standardised z-score.

In the first test, we permuted the timepoints while maintaining the treatment effects (*Dicer* KO vs control) to test the hypothesis that the observed phase delay differences in the stabilised circadian transcripts was not merely an outcome of altered expression levels (in particular lower amplitudes) in the *Dicer* KO (in which case it would be observable rather frequently on randomly generated cyclic transcripts). In the second test, the genotype (*Dicer* KO/control) was permuted to test the hypothesis that the observed effects on both amplitudes and phase differences were not random.

## Gene Ontology term and miRNA target enrichment analysis

Statistical testing of enrichment of Gene Ontology terms within lists of genes was performed with MetaCore version 6.17 software suite (Thomson Reuters, New York City, NY). For enrichment analysis of miRNA targets within lists of genes, the 'Conserved Family Info' table from the TargetScanMouse database version 6.2 (*Lewis et al., 2005*) was used to create an association database between miRNA families and the genes from the current study. Using an in-house Python script, raw counts of targets and non-targets were extracted from this database for the genes included in list. Background list included all 11,841 genes from *Figure 2F*. The statistical significance of enrichment of targets for each miRNA family in the test list against the background list was tested via Fisher's exact test on the contingency tables created from the raw counts. Calculated p-values were adjusted for FDR using BH method (*Benjamini and Hochberg, 1995*). Analysis was carried out in R environment (*R Core Team, 2013*).

## Data deposition

The RNA-seq data set produced in this study has been deposited in the Gene Expression Omnibus (accession number GSE57313). Supplementary files 1 and 2 are available at http://www.unil.ch/cig/page102471.html and have been deposited in the Dryad Digital Repository (*Du et al., 2014*).

## Acknowledgements

We thank Keith Harshman and the staff of the Lausanne Genomics Technologies Facility for high-throughput sequencing support; Ioannis Xenarios and the staff of the Vital-IT computational facility for computational support; Ueli Schibler for encouragement in the early stages of the project; Silvia Ferreira and Maykel Lopes for technical assistance; and Daniel Metzger, Pierre Chambon, Brian Harfe, Cliff Tabin, and Joseph Takahashi for mouse strains. We are indebted to Géraldine Mang, Peggy Janich and Kyle Gustafson for valuable comments on the manuscript.

## Additional information

### Funding

| Funder | Grant reference number | Author |
| --- | --- | --- |
| Swiss National Science Foundation | PP00P3_128399 | David Gatfield |
| Fondation Pierre Mercier | | David Gatfield |
| Novartis Foundation for Biomedical Research | 11A31 | David Gatfield |
| Fondation Leenaards | | David Gatfield |
| SystemsX.ch | StoNets | David Gatfield |
| University of Lausanne | | Ngoc-Hien Du, David Gatfield |

The funders had no role in study design, data collection and interpretation, or the decision to submit the work for publication.

### Author contributions

NHD, DG, Conception and design, Acquisition of data, Analysis and interpretation of data, Drafting or revising the article; ABA, Conception and design, Analysis and interpretation of data, Drafting or revising the article; MDM, Acquisition of data

### Ethics

Animal experimentation: Animal studies were conducted in accordance with the regulations of the veterinary office of the Canton of Vaud and were performed under authorization VD2376 to DG.

## Additional files

### Major datasets

The following datasets were generated:

| Author(s) | Year | Dataset title | Dataset ID and/or URL | Database, license, and accessibility information |
| --- | --- | --- | --- | --- |
| Du NH, Arpat AB, De Matos M, Gatfield D | 2014 | Data from: MicroRNAs Shape Circadian Hepatic Gene Expression on a Transcriptome-Wide Scale | http://dx.doi.org/10.5061/dryad.cd726 | Available at Dryad Digital Repository under a CC0 Public Domain Dedication. |
| Du NH, Arpat AB, De Matos M, Gatfield D | 2014 | MicroRNAs Shape Circadian Hepatic Gene Expression on a Transcriptome-Wide Scale | http://www.ncbi.nlm.nih.gov/geo/query/acc.cgi?acc=GSE57313 | Publicly available at NCBI Gene Expression Omnibus. |

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
