## [Decision Letter]

Thank you for sending your work entitled “MicroRNAs Shape Circadian Hepatic Gene Expression on a Transcriptome-Wide Scale” for consideration at *eLife*. Your article has been favorably evaluated by a Senior editor, a Reviewing editor, and 3 reviewers, one of whom, Ravi Allada, has agreed to reveal his identity.

Please address the questions/comments outlined below:

1) This study is very well presented, and experiments were performed and analyzed with great care. The approach is excellent with the use of an inducible, liver-specific *Dicer* knock-out. Results are solid, and conclusions sound. There is not much we can see that can be improved in this excellent manuscript. One minor point of concern is the fact that there is only a 50% overlap of cycling transcripts between dicer KO mice and controls. This seems quite low, so we wonder whether the threshold for rhythmicity was not too low in these experiments. If the threshold is increased, does the overlap increase?

2) Only a few transcripts (from thousands) show substantial effects on phase and amplitude in *Dicer*-KO. Our concern is that this might be random statistical effects due to high numbers of analyzed time-series (in thousands of randomly computer-generated synthetic time-series, one would also find such examples just by chance). Therefore, the authors should independently validate some of their data presented in Figure 4 and Figure 5 using e.g., qPCR.

3) In Figure 3, the increase of *Per1* mRNA abundance in *Dicer*-KO is pretty high compared to all other clock genes. However, in the reporter assay (Figure 6) there is almost no effect on *Per1*-3'UTR in the *Dicer*-KO (in contrast to *Per2* and *Cry2*) – so, this is not “consistent with the in vivo data from RNA-seq” as the authors write – how can this be explained?

---

## [Author Response]

*1) This study is very well presented, and experiments were performed and analyzed with great care. The approach is excellent with the use of an inducible, liver-specific* Dicer *knock-out. Results are solid, and conclusions sound. There is not much we can see that can be improved in this excellent manuscript. One minor point of concern is the fact that there is only a 50% overlap of cycling transcripts between dicer KO mice and controls. This seems quite low, so we wonder whether the threshold for rhythmicity was not too low in these experiments. If the threshold is increased, does the overlap increase?*

We have performed the suggested analysis by varying three main parameters used for rhythmicity detection: False Discovery Rate (FDR, default 5 %, now decreased to 1 % for higher threshold), amplitude cut-off (default 1.5 x, now altered to 2 x for higher threshold) and the methods used (default: rhythm detection by “JTK_cycle OR harmonic fit”; now used as “JTK_cycle AND harmonic fit”). As shown in Figure 4—figure supplement 2, varying these parameters did not lead to improved overlap between the genotypes. On the contrary: under very high stringency conditions (Figure 4—figure supplement 2) the overlap became even smaller.

This observation led us to conclude that the poor overlap was probably not a technical issue, but could rather be indicative of a robust difference between the genotypes that had a biological basis. The most intriguing aspect of the poor overlap was the fact that many transcripts appeared to become rhythmic in the Dicer KO (see Figure 4, sectors B-D). We used GO term analysis (Table S3) to obtain an idea of the biological basis and visual inspection of the affected genes to confirm the findings. Interestingly, these analyses showed that there were indeed many transcripts that – transcriptionally and post-transcriptionally – became rhythmic specifically in the *Dicer* knockouts. Many of these had connections to the cell cycle and DNA replication (e.g., the Mcm complex), showed high expression and high amplitude rhythms in the knockouts, but frequently lower expression in the controls. We have assembled a number of examples in Figure 4—figure supplement 3. In the initial version of the manuscript (in connection with Figure 1—figure supplement 1) we had speculated about increased cellular turnover rates in the *Dicer* knockouts. The synchronised expression of cell cycle and DNA replication genes confirms this initial suspicion and is in line with findings on the circadian gating of the cell cycle in the regenerating liver that was demonstrated by the Okamura lab 10 years ago (Matsuo et al. Science 2003). We have integrated the figure supplements in the main manuscript and describe the findings in the revised version.

*2) Only a few transcripts (from thousands) show substantial effects on phase and amplitude in* Dicer*-KO. Our concern is that this might be random statistical effects due to high numbers of analyzed time-series (in thousands of randomly computer-generated synthetic time-series, one would also find such examples just by chance). Therefore, the authors should independently validate some of their data presented in*
Figure 4
*and*
Figure 5
*using e.g., qPCR*.

We have performed the requested qPCR analyses on a selection of transcripts (mRNA and pre-mRNA) from the different categories/Figures and we have added the data as Figure 4—figure supplement 5. The qPCRs thus confirmed the specific expression profiles detected by RNA-seq. However, for many obvious reasons, qPCR is inferior to RNA-seq in terms of specificity, sensitivity and noise. Especially for low abundance pre-mRNAs, the results are more variable than the plots shown in the main Figure 4 and 5E-H for RNA-seq.

Moreover, whereas the reviewers’ concern about whether “*substantial effects on phase and amplitude […] might be random statistical effects due to high numbers of analyzed time-series (in thousands of randomly computer-generated synthetic time-series, one would also find such examples just by chance)*” is a very valid question, we are not sure if the qPCRs are the best and most appropriate way of addressing it.

We have thus performed an additional dedicated analysis of whether the correlations we observed could potentially be explained by random statistical effects. To this end, we have implemented two permutation tests (N>10000) on the original data. Each permuted dataset was then analysed by exactly the same steps used on the experimental data. Finally, the distribution of the difference in means, significance of the effects and correlation coefficients was inspected for both amplitude and phase effect, and the results of the permutations were then compared to the original data.

In the first test, we permuted the timepoints while maintaining the genotypes (*Dicer* KO/control) to evaluate whether the observed phase delay differences for the stabilised circadian transcripts (higher mRNA/pre-mRNA ratio) was merely an outcome of altered expression levels in the *Dicer* KO or of some other feature in the analysis pipeline (in which case it should be observable rather frequently on randomly generated cyclic transcripts). However, as shown in Figure 5—figure supplement 2, our reported phase delay phenotype was a rare event in the random data in terms of the magnitude of the effect (Figure 5—figure supplement 2, left panel: only 2.8 % of permutations showed a similar or greater delay), its statistics calculated by the Wilcoxon test (not a single permutation was as significant) and the Pearson’s correlation between phase delay differences and mRNA/pre-mRNA ratios (4.7 % of permutations did equally well or better). Since in this permutation analysis the genotypes remained unchanged, the effect of dampened amplitudes in the knockout were retained and were quite frequently reproduced in the permutation data as well (Figure 5—figure supplement 2).

Therefore, in the second test, the genotypes were permuted (but timepoints were not) to test the hypothesis that the observed effects on both amplitudes and phase differences could have been not random. However, both for phase and amplitude effects, similar (or stronger) outcomes were observed extremely rarely in the random time series (Figure 5—figure supplement 2).

In summary, we thus concluded that the observed phenotypes of both amplitude dampening and phase delay differences were not random and not due to, for example, a specific bias in the RNA-seq count data that would predispose to such outcomes. We have integrated the data into the manuscript and discuss these analyses in the Results and Materials and Methods sections of the revised version.

*3) In*
Figure 3*, the increase of* Per1 *mRNA abundance in* Dicer*-KO is pretty high compared to all other clock genes. However, in the reporter assay (*Figure 6*) there is almost no effect on* Per1*-3'UTR in the* Dicer*-KO (in contrast to* Per2 *and* Cry2*) – so, this is not “consistent with the in vivo data from RNA-seq” as the authors write – how can this be explained?*

We agree with the reviewers and apologise for the lack of precision regarding the description of the *Per1* data. Indeed, the result of the *Per1*-3’ UTR reporter assay is consistent with the western blot in Figure 3 (in both cases ca. 1.1-1.2 fold increase in *Dicer* KO) but not with the *Per1* mRNA data, which shows a stronger effect (factor 2 x). The main discrepancy is therefore at the level of the two in vivo measures in Figure 3 (*Per1* mRNA) and 3C (PER1 protein). We have now reworded the corresponding passages in order to avoid potential misunderstandings. Moreover, we have added the values for western blot quantification of PER2, PER1 and CRY2 in Figure 3 in order to demonstrate the generally good agreement between in vivo protein data and the luciferase assays – with the notable exception of PER1.

A number of mechanisms could account for the observed uncoupling of protein from mRNA levels, including secondary effects of miRNA loss acting on *Per1* mRNA translation or the protein degradation machinery. While in this manuscript we do not have the means to conclude on the exact mechanism(s) that is (are) operative in counter-regulating the *Per1* mRNA increase on the protein level, it is interesting to note that there is an RNA binding protein, hnRNP Q/Syncrip, which has been reported to enhance the translation of *Per1* mRNA (64). In the *Dicer* knockout, *hnRNP Q* mRNA shows decreased expression (Figure 3—figure supplement 1), which could thus lead to decreased PER1 translation and protein abundance, hence counteracting increased mRNA levels.

However, this case is likely even more complicated, because the aforementioned work on hnRNP Q (64) suggests that this protein acts through sequences that are located at the 5’ end of the *Per1* mRNA. This mechanism could therefore contribute to regulating translational efficiencies for endogenous *Per1*, but not for the luciferase reporter, which only carries the *Per1* 3’ UTR.

We have nevertheless decided to mention this example in the manuscript, as it illustrates the type of mechanisms that would need to be operative for the mRNA-protein uncoupling. It would require substantial additional work to define the precise molecular mechanisms responsible for the interesting observation of *Per1* mRNA-protein uncoupling.